# Geolocator tagging links distributions in the non-breeding season to population genetic structure in a sentinel North Pacific seabird

J. Mark Hipfner[1]*, Marie M. Prill[2], Katharine R. Studholme[3], Alice D. Domalik[4¤a], Strahan Tucker[5], Catherine Jardine[6], Mark Maftei[1¤b], Kenneth G. Wright[1], Jessie N. Beck[7], Russell W. Bradley[8¤c], Ryan D. Carle[7], Thomas P. Good[9], Scott A. Hatch[10], Peter J. Hodum[11], Motohiro Ito[12], Scott F. Pearson[13], Nora A. Rojek[14], Leslie Slater[14], Yutaka Watanuki[15], Alexis P. Will[16], Aidan D. Bindoff[17], Glenn T. Crossin[3], Mark C. Drever[1], Theresa M. Burg[2]

1 Wildlife Research Division, Environment and Climate Change Canada, Delta, British Columbia, Canada, 2 Department of Biology, University of Lethbridge, Lethbridge, Alberta, Canada, 3 Department of Biology, Dalhousie University, Halifax, Nova Scotia, Canada, 4 Department of Biological Sciences, Simon Fraser University, Burnaby, British Columbia, Canada, 5 Fisheries and Oceans Canada, Nanaimo, British Columbia, Canada, 6 Birds Canada, Delta, British Columbia, Canada, 7 Oikonos Ecosystems Knowledge, Santa Cruz, California, United States of America, 8 Point Blue Conservation Science, Petaluma, California, United States of America, 9 Northwest Fisheries Science Centre, National Marine Fisheries Service, National Oceanographic and Atmospheric Administration, Seattle, Washington, United States of America, 10 Institute for Seabird Research and Conservation, Anchorage, Alaska, United States of America, 11 Department of Biology, University of Puget Sound, Tacoma, Washington, United States of America, 12 Department of Applied Biosciences, Toyo University, Bunkyō-ku, Japan, 13 Washington Department of Fish and Wildlife, Olympia, Washington, United States of America, 14 United States Fish and Wildlife Service, Homer, Alaska, United States of America, 15 Graduate School of Fisheries Sciences, Hokkaido University, Hakodate, Hokkaido, Japan, 16 Institute of Arctic Biology, University of Alaska Fairbanks, Fairbanks, Alaska, United States of America, 17 Wicking Dementia Research and Education Centre, University of Tasmania, Hobart, Tasmania, Australia

¤a Current address: Wildlife Research Division, Environment and Climate Change Canada, Delta, British Columbia, Canada
¤b Current address: Raincoast Education Society, Tofino, British Columbia, Canada
¤c Current address: California State University Channel Islands, Camarillo, California, United States of America
* mark.hipfner@canada.ca

**Data Availability Statement:** The data that support the findings of this study are available on Dryad (microsatellites) https://datadryad.org/stash/share/

## Abstract

We tested the hypothesis that segregation in wintering areas is associated with population differentiation in a sentinel North Pacific seabird, the rhinoceros auklet (*Cerorhinca monocerata*). We collected tissue samples for genetic analyses on five breeding colonies in the western Pacific Ocean (Japan) and on 13 colonies in the eastern Pacific Ocean (California to Alaska), and deployed light-level geolocator tags on 12 eastern Pacific colonies to delineate wintering areas. Geolocator tags were deployed previously on one colony in Japan. There was strong genetic differentiation between populations in the eastern vs. western Pacific Ocean, likely due to two factors. First, glaciation over the North Pacific in the late Pleistocene might have forced a southward range shift that historically isolated the eastern and western populations. And second, deep-ocean habitat along the northern continental shelf appears to act as a barrier to movement; abundant on both sides of the North Pacific, the rhinoceros auklet is virtually absent as a breeder in the Aleutian Islands and Bering Sea,

I_CbHxHbpF4KFARdQ2OqPDzRo9SpfAc9I
QanRgZUk7Y and on Movebank (GLS tracking
data) https://www.movebank.org/cms/webapp?
gwt_fragment=page=studies,path=study12518
47150.

**Funding:** Primary funding was provided by
Environment and Climate Change Canada (J.M.H.),
the Natural Sciences and Engineering Research
Council of Canada (Discovery Grants to G.T.C. and
T.M.B.), the North Pacific Research Board
(Graduate Student Research Awards to K.R.S.
(2014) and A.P.W. (2015)) the United States Fish
and Wildlife Service, the Alaska Maritime National
Wildlife Refuge (N.R. and L.S.), the Farallon Islands
National Wildlife Refuge (R.W.B.), the Washington
State Department of Fish and Wildlife (S.F.P.), and
a Japan Society for the Promotion of Science
Grant-in Aid for Scientific Research (16H06739 and
17K15308 to M.I.). The funders played no role in
study design, data collection and analysis, decision
to publish, or preparation of the manuscript.

**Competing interests:** The authors have declared
that no competing interests exist.

and no tagged birds crossed the North Pacific in the non-breeding season. While genetic differentiation was strongest between the eastern vs. western Pacific, there was also extensive differentiation within both regional groups. In pairwise comparisons among the eastern Pacific colonies, the standardized measure of genetic differentiation ($F'_{ST}$) was negatively correlated with the extent of spatial overlap in wintering areas. That result supports the hypothesis that segregation in the non-breeding season is linked to genetic structure. Philopatry and a neritic foraging habit probably also contribute to the structuring. Widely distributed, vulnerable to anthropogenic stressors, and exhibiting extensive genetic structure, the rhinoceros auklet is fully indicative of the scope of the conservation challenges posed by seabirds.

## Introduction

Barriers to dispersal are generally thought to be less rigid and absolute in marine environments than in terrestrial environments, implying that genetic connections in marine organisms will span longer distances, and population genetic structuring will be less prevalent [1–3]. Many marine organisms also exist at very high densities, which reduces the potential for genetic differentiation to arise from drift [4]. Nonetheless, a wide variety of effectively unbridgeable barriers to dispersal and gene flow, many of them cryptic, are present in the ocean. These barriers include spatial gradients in salinity [5] and temperature [6], eddies and gyres [7], fronts [8], ocean currents [9], irregular coastlines [10], and weak connections between ocean basins [11]. Consequently, population genetic structure exists in a broad range of marine taxa, including phytoplankton [12], vascular plants [13], crustacean zooplankton [14], molluscs [15], cephalopods [16], and vertebrates including marine fish [17], reptiles [18], and mammals [19]. Interpreting the complex patterns of genetic structuring displayed by marine organisms is a task that continues to challenge conservation researchers and managers [4, 20], suggesting the utility of targeted, hypothesis-driven approaches [21].

Among marine vertebrates, seabirds possess a suite of ecological and behavioural traits that are particularly relevant to investigations of population differentiation [22, 23]. A taxonomically diverse group usually considered to include ~360 species in six Orders, seabirds inhabit all of the world's oceans and share a core suite of "slow" life-history traits: long pre-breeding periods, small clutch sizes, and high adult survival rates [24, 25]. Among the traits common to seabirds that are especially germane to population genetics are extreme mobility, philopatry, and high-density breeding in discrete groups (colonies). Most seabirds are strong fliers capable of rapidly traversing long stretches of ocean. However, the variation in dispersal capability across the group is considerable, with the flightless penguins (all species) and cormorants (one species) at one extreme, and the albatrosses, wide-ranging ocean wanderers [26], at the other. Such high capacity for dispersal violates the assumptions of many models of population divergence and speciation [22], and may render some of the smaller-scale oceanographic barriers that limit gene flow in other marine taxa irrelevant to seabirds. Conversely, philopatry restricts gene flow [27], thereby promoting population structure for species that return annually to the same breeding colonies. From a logistical perspective–spatial and temporal predictability, ease of access, sample sizes–the habit of colonial breeding generally makes seabirds good subjects for study [23].

Friesen, Burg, & McCoy [22] reviewed the literature on the extent and causes of population genetic structuring in seabirds based on mitochondrial DNA. They found that among all species studied, the extent of structuring varied from virtual panmixia to reciprocal monophyly,

but structure was present in most. More specifically, genetic or genetic-plus-phylogeographic structure was found in all 12 species with breeding distributions fragmented by rigid physical barriers (land and/or ice), and in 26 of 37 species (70%) with continuous distributions. From among a suite of seven factors proposed to explain structuring in the latter group (geographic distance, the pattern of colony dispersion, distribution outside the breeding season, foraging range around colonies, population bottlenecks, retained ancestral variation, and cryptic physical barriers), the presence or absence of population genetic structure appeared to be best explained by distributions outside the breeding season. Structure was associated with the behavioural tendency to occupy multiple population-specific non-breeding areas, and/or to reside year-round at or near breeding colonies.

Two more recent reviews have further assessed the factors underlying genetic structuring in seabirds based on mitochondrial and nuclear DNA. The first of these reviews [27] concluded that geographic barriers, natal philopatry, and the occupation of multiple oceanic regions (e.g., basins, current systems) were important factors restricting gene flow. The second review [28] examined seabirds of the Southern Ocean, a marine region relatively free of physical barriers to dispersal [29], and identified ocean currents (especially for penguins) and philopatry as primary factors promoting structure. Of note, both of the newer studies confirmed one of the major conclusions of the original [22], specifically, that spatial separation in the non-breeding season was associated with restricted gene flow. Surprisingly, given its prominence in the reviews, the hypothesis is rarely tested directly. In two investigations to date, one found no support in populations of three species of seabirds that exhibited spatio-temporal segregation at sea, but little genetic structure [30]; and the second found support for the hypothesis, but concluded that gene flow between the two populations being investigated was restricted not by spatial segregation in the non-breeding season *per se*, but by population-level differences in the timing of breeding linked to habitat specialization at that time [31].

Here, we test the hypothesis that segregation in wintering areas is associated with genetic differentiation in the rhinoceros auklet, a widely-distributed North Pacific seabird that exhibits extensive and complex population genetic structuring [32]. The migratory movements of the species in the eastern Pacific are poorly known, however, beyond a putative southward post-breeding trajectory, e.g., "the bulk of the population appears to winter off California" [33]; "winters mostly offshore and along coasts, in North America mainly from southern British Columbia (casually from SE Alaska) south to Baja California" [34]. Our approach was to expand on the study of Abbott et al. [32] by increasing sample sizes for microsatellites, individuals, and colonies, while concurrently deploying light-level geolocator (GLS) tags to track the migration of rhinoceros auklets from 12 colonies distributed more-or-less continuously along the Pacific Ocean coastline from California to the Alaska Peninsula. For all colony-pairs, we then correlated the standardized measure of genetic differentiation ($F'_{ST}$) with the extent of spatial overlap in wintering areas. We also tested for isolation by distance (IBD) among the eastern Pacific Ocean colonies, to evaluate whether average dispersal distance accounts for genetic structure, assuming all populations have equal mean dispersal distances. Finally, we collected blood samples on five colonies in Japan, including one site where geolocator tags were deployed previously [35]. Our sampling thus spanned the global range of the species, the only exception being the Russian Far East, which was found to support large numbers in recent surveys [36].

## Materials and methods

### Study species, study locations, and oceanographic setting

The rhinoceros auklet is an abundant, colonial, burrow-nesting seabird with a wide distribution across the temperate North Pacific Ocean. The species is misnamed, in that it does not

belong to the auklet clade (Aethiini) within the family Alcidae. Rather, the genus *Cerorhinca* is either basal within the puffin clade (Fraterculini), or it forms a sister clade with *Fratercula* [37].

The main part of our study occurred on 13 rhinoceros auklet breeding colonies in the eastern North Pacific Ocean (Fig 1). These consisted of three colonies in Alaska (Chowiet, Middleton, and St. Lazaria islands), six colonies in British Columbia, which supports the majority of the North American breeding population (Lucy, Moore, Pine, Triangle, and Cleland islands, plus S'Gang Gwaay), two colonies in Washington State (Protection and Destruction islands), and two colonies in California (Southeast Farallon and Año Nuevo islands). The North Pacific Current flows east from Asia, bifurcating roughly off southern British Columbia into the southward-flowing California Current, an upwelling system, and the northward-flowing Alaska Current, a downwelling system [38]. These two systems exert the major oceanographic influences around the 13 study colonies. From a historical perspective, many of the colonies north of California would have been covered by the Cordilleran Ice Sheet at its maximum extent ca. 19–14.5 kya near the end of the Pleistocene [39, 40].

For the genetics we also included five western North Pacific colonies, all in Japan: three in the Japan Sea off Hokkaido (Todojima, Teuri, Matsumae-Kojima Islands), one along the Pacific coast of Hokkaido (Daikoku Island), and one in Mutsu Bay in Aomori (Taijima Island). Geolocator tags were deployed previously on rhinoceros auklets on Teuri Island [35]. The location and population size of all 18 colonies are provided in Table 1.

## Genetics methods

Research protocols employed in this study were approved by Simon Fraser University Animal Care Services (#974B-94), the Western and Northern Animal Care Committee of Environment and Climate Change Canada's Canadian Wildlife Service (15MH01 and 16MH01), US Geological Survey Federal Bird Banding Permit (#09316, #20570), the Animal Ethics Committee of Hokkaido University, and the Aomoru prefecture (#3021). Access to field sites was provided by the Año Nuevo State Park, the California Department of Parks and Recreation, the United States Fish and Wildlife Service, the Washington Department of Fish and Wildlife, British Columbia Parks, the Archipelago Management Board of Gwaii Haanas National Park Reserve, and the Japanese Agency for Cultural Affairs.

Rhinoceros auklets involved in the genetics (2010–2016) and geolocator tagging (2014–2016) studies were captured using a variety of methods. Most were removed from breeding burrows, others blocked while departing the colony in knock-down or mist nets, others caught in purse nets or noose mats set in the entrances of burrows, and others captured by hand on the surface.

Blood samples (1.0 ml) and/or feather samples (5–10, from the breast) were collected from seven to 80 individuals on each of the 18 colonies, for a total of 704 individuals. All samples collected for the earlier study [32] were included, and supplemented with additional samples from Chowiet and St. Lazaria islands, and from 10 new sites. Samples were stored in ethanol and then at -20°C upon return to the lab. DNA was extracted from samples collected at four sites in Japan (Daikoku, Taijima, Matsumae-Kojima, Todojima) using the DNeasy® Blood and Tissue Kit (Qiagen), and at one of our California sites (Año Nuevo, where muscle tissue from nestlings found dead in burrows was used) using Macherey-Nagel DNA extraction kit. For all remaining samples, DNA was extracted using a modified Chelex protocol [47, 48].

A small set of samples (three to six) was genotyped to check for amplification and polymorphism with microsatellite loci from the genomes of rhinoceros auklet, crested auklet (*Aethia cristatella*), and whiskered auklet (*Aethia pygmaea*). Of the 31 microsatellite loci tested, 12

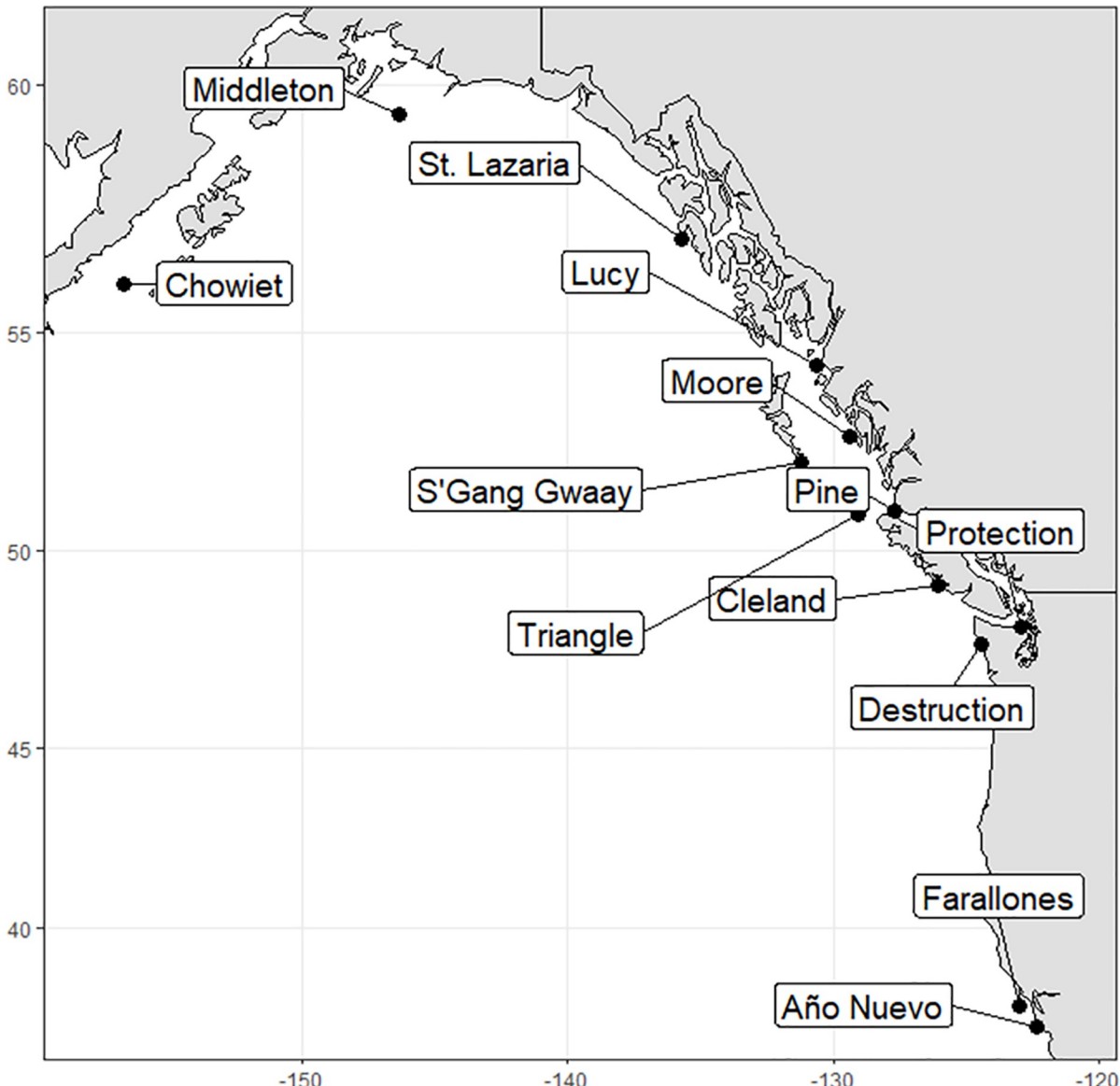

**Fig 1. Locations of rhinoceros auklet breeding colonies along the Pacific Coast of North America where samples were collected for genetics analysis and light-level geolocator tags were deployed.**

were polymorphic: CMms2, CMms3, CMms4, CMms9, CMms14, CMms22, CMms23, and CMms26 [49]; Pal11 and Pal26 [50]; Apy06 and Apy09 [51]. The eight CMms loci were the same ones used in the earlier study [32]. Due to inconsistent amplification, Pal26 and Apy09 were removed from the study and individuals were genotyped at the remaining ten microsatellite loci.

DNA was amplified in a 10 μL polymerase chain reaction (PCR) containing colorless GoTaq® Flexi (Promega) or Truin buffer, 0.2 mM dNTP, 2.5 mM MgCl$_2$, 0.5 μM forward and 1 μM reverse primers, 0.05 μM M13 fluorescently labelled primer and 0.5 U GoTaq® Flexi or 1 U Truin Taq polymerase. The lone exception was CMms4, for which 3 mM MgCl$_2$ was used. One percent formamide was added to CMms4, CMms9, and CMms14 reactions. A M13 sequence was added to the 5' end of the forward primer to allow incorporation of a

**Table 1. Locations of rhinoceros auklet breeding colonies included in the genetics study, and the sizes of their breeding populations.**

| Breeding colony | Location | Population size (pairs) | Latitude (˚N) | Longitude (˚W) |
|---|---|---|---|---|
| Daikoku | Japan | 78,000 [41] | 42.954744 | 144.866057 |
| Taijima | Japan | 150 [41] | 41.263284 | 140.345585 |
| Matsumae-Kojima | Japan | 40,000 [41] | 41.360155 | 139.818113 |
| Teuri | Japan | 379,000 [41] | 44.417646 | 141.312281 |
| Todojima | Japan | 35,000 [41] | 45.366808 | 141.035141 |
| Chowiet | Alaska | 400 [42] | 56.015513 | -156.740272 |
| Middleton | Alaska | 10,000 (S. Hatch, unpublished data) | 59.415254 | -146.345472 |
| St. Lazaria | Alaska | 2,000 [43] | 56.986502 | -135.710838 |
| Lucy | British Columbia | 25,000 [44] | 54.294418 | -130.621907 |
| S'Gang Gwaay | British Columbia | 14,000 [44] | 52.092634 | -131.225633 |
| Moore | British Columbia | 40,000 [44] | 52.678344 | -129.418847 |
| Triangle | British Columbia | 42,000 [44] | 50.851023 | -129.066292 |
| Pine | British Columbia | 90,000 [44] | 50.976062 | -127.729909 |
| Cleland | British Columbia | 1,000 [44] | 49.171516 | -126.091075 |
| Destruction | Washington | 6,500 [45] | 47.674599 | -124.484817 |
| Protection | Washington | 36,000 [45] | 48.126341 | -122.930289 |
| Southeast Farallon | California | 4,500 [46] | 37.695357 | -123.000752 |
| Año Nuevo | California | 330 [46] | 37.107584 | -122.337026 |

fluorescently-labelled M13 primer for visualization of the PCR product. The Apy09 reverse primer was pig-tailed (gtttctt) at the 5' end to improve scoring [52]. Nine loci were amplified using the following thermal cycling profile: 120 s at 94˚C, 45 s at 55˚C, and 60 s at 72˚C; seven cycles of 60 s at 94˚C, 30 s at 55˚C, and 45 s at 72˚C; 31 cycles of 30 s at 94˚C, 30 s at 57˚C and 45 s at 72˚C; and a final cycle at 72˚C for 300 s. The tenth locus, Pal11, was amplified using the same program except the steps annealing $T_{a1}$ and $T_{a2}$ were increased to 60˚C and 62˚C respectively.

PCR products were run on a 6% polyacrylamide gel using a LI-COR 4300 DNA Analyzer (LI-COR Inc.). Alleles were scored via visual inspection with all genotypes confirmed by a second person. To maintain consistent scoring, three positive controls of known allele size were present on each load.

## Genetic diversity analyses

Seven to 53 individuals remained from each breeding colony after excluding individuals missing three or more loci for a total of 424 individuals from 18 breeding colonies genotyped with ten microsatellites loci (Table 2). For the seven individuals from the Southeast Farallon colony, CMms3 failed to amplify. CMms26 had a high percentage of missing data (> 35%). Thus, analyses sensitive to missing data excluded the colony at Southeast Farallon and the CMms26 locus. GENEPOP v4.2 [53, 54] was used to check colonies and loci for linkage disequilibrium and deviations from Hardy-Weinberg equilibrium (HWE) using Markov chain parameters of 1000 iterations, 300 batches, and 2,000 dememorization steps. MICRO-CHECKER v2.2.3 [55] was used to detect scoring error due to stutter, null alleles, and drop out of large alleles.

Levels of population genetic diversity, observed and expected heterozygosities, private alleles, and number of alleles per locus were calculated in GenAlEx v6.5 [56, 57]. Because smaller sample sizes are expected to have fewer alleles, allelic richness was calculated using statistical rarefaction in HP-Rare v1.1 [58] which standardizes measurements to account for differences in sample size.

**Table 2. Rhinoceros auklet breeding colonies (populations) included in the genetics analyses, sample size (*n*), number of alleles (Na), expected heterozygosity (H$_E$), observed heterozygosity (H$_O$), private alleles (P$_A$), and allelic richness (A$_R$).**

| Breeding colony | *n* | Na | H$_E$ | H$_O$ | P$_A$ | A$_R$ |
|---|---|---|---|---|---|---|
| Daikoku | 20 | 5.3 | 0.65 | 0.68 | 1 | 3.89 |
| Taijima | 10 | 4.8 | 0.65 | 0.77 | 0 | 3.97 |
| Matsumae-Kojima | 25 | 5.3 | 0.64 | 0.71 | 0 | 3.76 |
| Teuri | 21 | 5.4 | 0.63 | 0.66 | 1 | 3.61 |
| Todojima | 12 | 4.6 | 0.64 | 0.67 | 0 | 3.75 |
| Chowiet | 18 | 4.9 | 0.59 | 0.59 | 0 | 3.58 |
| Middleton | 50 | 6.3 | 0.64 | 0.60 | 1 | 3.71 |
| St. Lazaria | 22 | 5.9 | 0.65 | 0.65 | 1 | 3.87 |
| Lucy | 26 | 5.3 | 0.60 | 0.63 | 0 | 3.53 |
| S'Gang Gwaay | 53 | 5.5 | 0.62 | 0.61 | 1 | 3.56 |
| Moore | 18 | 4.8 | 0.62 | 0.62 | 0 | 3.55 |
| Triangle | 26 | 4.7 | 0.61 | 0.62 | 0 | 3.44 |
| Pine | 27 | 5.6 | 0.63 | 0.60 | 0 | 3.64 |
| Cleland | 30 | 5.8 | 0.66 | 0.64 | 1 | 3.85 |
| Destruction | 18 | 4.9 | 0.66 | 0.71 | 0 | 3.94 |
| Protection | 19 | 5.2 | 0.61 | 0.63 | 0 | 3.71 |
| Southeast Farallon | 7 | 3.3 | - | - | - | - |
| Año Nuevo | 22 | 5.5 | 0.66 | 0.68 | 1 | 3.83 |

For some measures, values are excluded for Southeast Farallon Island because of small sample size and data missing from locus CMms3.

GenAlEx v6.5 [56, 57] was used to calculate both global and pairwise F$_{ST}$ and F'$_{ST}$ values to characterise population genetic structure with 999 permutations used to test significance. To address the multi-allelic nature of microsatellites, F'$_{ST}$ provides a standardized value by dividing each F$_{ST}$ with the maximum possible F$_{ST}$ for the data [59]. For all tests with multiple comparisons, statistical levels of significance were adjusted using the modified False Discovery Rate (FDR) correction [60].

STRUCTURE v2.3.4 [61, 62], a non-spatial Bayesian method, was used to determine the number of genetic clusters present among populations. STRUCTURE captures the underlying population structure of the data without overestimating it. However, the program struggles to cluster individuals when genetic differentiation is low (F$_{ST}$ ≤ 0.03) [63] and substructure might be present among populations. Therefore, we applied a hierarchical STRUCTURE approach (i.e., separate multistep runs with admixed individuals) that might detect genetic structure that is not apparent when all populations are run together [64].

STRUCTURE v2.3.4 was run using the recommended *admixture* ancestry model and *correlated* allele frequencies with sampling locations as *locpriors*. The *locpriors* option can be informative when population structure is weak [65]. For the 18 breeding colonies, ten independent runs were completed for each value of K from 1–6 with a burn-in of 100,000 and 120,000 Markov chain Monte Carlo (McMC) repetitions. The most appropriate number of clusters (K) was determined using several methods including lnPr(X|K) values, STRUCTURE HARVESTER v0.6.94 [66] and Bayes factor [62]. To further measure whether substructure was present within the populations, a hierarchical analysis for both the western (n = 5) and eastern (n = 13) breeding groups was completed using a burn-in of 60,000 and 70,000 McMC repetitions. K

ranged from K = 1–4 for the western group and K = 1–5 for the eastern group. Five additional runs were completed at the optimal K to ensure convergence.

GenAlEx v6.5 [56, 57] was used to perform a principal coordinates analysis (PCoA) using standardized covariance from the $F'_{ST}$ pairwise matrix. Three separate analyses were performed: all colonies, only western Pacific colonies, and only eastern Pacific colonies.

## Geolocator deployments

We deployed light-level based Global Location Sensing (GLS) tags (Intigeo-C65, Migrate Technology Ltd., 1.0 g, range 4, mode 6) on breeding rhinoceros auklets on 13 eastern Pacific colonies in 2014 and 2015 (Fig 1). Most tags were deployed during the second half of the offspring provisioning period, as close to fledging as possible. Each tag was pre-attached to a Darvic leg band using self-amalgamating tape, epoxy, and an ultraviolet-resistant zip tie with a stainless-steel barb threaded through two custom holes in the band such that the whole assembly (< 2.0 g) could be quickly attached to the tarsus. Tags were groundtruthed (run at a known location in the colony) for ≥ 3 days prior to deployment. At deployment, the morphometrics of each bird were recorded, a numbered USGS metal band and a geolocator tag assembly were applied to the right and left tarsi, respectively, and the bird was released. Efforts were made to recapture birds and retrieve the tags beginning early in the following breeding season. Retrieved tags were groundtruthed again for ≥4 days. It was not possible to test for tag-induced changes to non-breeding behavior, and because birds can change burrows or nesting areas among years, failure to recapture a bird during this study does not necessarily indicate mortality.

## Geolocation data analyses

We used the TwilightFree package to derive daily locations from the light level data [67]. This approach uses a Hidden Markov Model in which the hidden states are the daily geographic locations and the measured response is the observed pattern of light and dark over a 24 h period, and identifies the most likely tracks (based on the centre coordinates of a pre-defined grid) predicted to have generated the light record. The user specifies the spatial grid, minimum light threshold, zenith, and parameters related to the probability of light sensor shading and of movements between grid cells [67].

Based on the ground-truthing periods, a preliminary calibration was conducted on each light record to determine the minimum light threshold and zenith, which were used for all runs (zenith = 96˚, threshold = 10 lux). We defined a grid that spanned from -180 to -100˚ longitude, and 10 to 65˚ latitude, and used a sea mask to restrict movements to the ocean. Shading likelihood parameters were selected to represent moderate amounts of shading (alpha = 5), and movement probabilities based on other seabirds (beta = 5) [67]. Each lux file was then processed wherein the model was initially fit with a grid of 4˚×4˚ cell sizes to identify days with missing light data, which were then excluded from subsequent analyses. This reduced light record was then refit on a finer grid of 1˚×1˚ cell sizes, and the track files derived from this second fit used in further analyses.

**Home ranges in winter.** The breeding seasons of rhinoceros auklets on colonies in the eastern Pacific Ocean span the period from April to August, generally starting and ending about one month earlier in the south than in the north [33]. In defining a non-breeding season for this study, we restricted tracks to the period from 1 November to 28/29 February. These dates roughly span the period from the end of fall migration (post-breeding dispersal from the colony) to the start of spring migration (pre-breeding return to the colony). These dates also

avoid errors in geolocation due to the confounding of movements and light levels that occurs at the equinox.

We characterized the non-breeding distribution of each bird by its Utilization Distribution (UD), the bivariate function giving the probability density that an animal is found at a point in space according to its geographical coordinates [68]. The daily spatial coordinates were used to calculate the UD for each individual bird, as well as an aggregate from all the tags deployed in each colony. The two years of deployment (2014, 2015) were initially considered separately, and then were consolidated following the observation that tags from each colony had very high overlaps across years. We used function kernelUD from package adehabitatHR in R [69] to calculate the UDs, with the *ad hoc* method to derive the smoothing parameter and estimated over a grid of 1˚×1˚ cell sizes. The function getverticeshr was used to extract the 50 and 90 percent home-range contours for each bird or colony aggregate. These contours were projected to a Lambert Equal-Area Azimuthal projection, and the area of the polygons was calculated in km2.

**Spatial overlap.** The function kerneloverlaphr was used to calculate the UD overlap index (UDOI) between each pairwise combination of colony UDs [70]. The UDOI is 1.0 when both UDs are uniformly distributed and share 100% overlap, 0 when UDs have no overlap, and > 1.0 when the two UDs are non-uniformly distributed and share high overlap.

**Correlations with F'st.** To test whether genetic distance between rhinoceros auklets from different breeding colonies was related to (i) the distance between breeding colonies, or (ii) the spatial overlap among colonies in winter, we correlated these two measures with the standardized measure of genetic differentiation (F'st) values based on Wright's fixation indices. We calculated Spearman's correlation (r) between F'st values and (i) distance between breeding colonies (IBD) or (ii) the UDOIs. Given the lack of independence between pairwise values, we calculated the statistical significance of r-values using a randomization procedure similar to a Mantel test. We built a sampling distribution of r through a permutation of holding F'st values and randomly assigning it a pair from the available UDOI or IBD values. This was repeated 5000 times, and the proportion of times when the randomized r was larger than the observed r was used as an approximate P-value. Given the variability in the underlying data, we used an alpha-level of 0.10 to minimize the probability of a Type II statistical error.

# Results

## Genetic diversity analyses

Over all colonies and loci, the number of alleles ranged from 2 to 16 with seven colonies having one private allele. Excluding the Southeast Farallon colony due to small sample size and missing data from locus CMms3, overall mean expected heterozygosity across all loci and samples was 0.62. Expected heterozygosity ranged from 0.59 (Chowiet) to 0.66 (Cleland, Destruction and Año Nuevo) with observed heterozygosity 0.59 (Chowiet) to 0.77 (Taijima). Allelic richness (corrected to a sample size of 10) was similar among colonies, ranging from 3.44–3.97 (Table 2). After FDR correction, two loci (CMms2, CMms22 at St. Lazaria) showed deviations from HWE (Table 2). There was no evidence for linkage disequilibrium between any of the loci. MICRO-CHECKER found no evidence of null alleles, large allele dropout or scoring error due to stutter.

$F_{ST}$ statistics excluded the Southeast Farallon colony and CMms26 locus because of small sample sizes and missing data, respectively. Using 17 colonies and nine microsatellite loci, global $F_{ST}$ was 0.039 ($p < 0.001$). Pairwise $F_{ST}$ values ranged from 0.000 (Pine and Chowiet) to 0.112 (Lucy and Teuri), and F'$_{ST}$ values ranged from 0.000 to 0.307 (Table 3). After FDR

**Table 3. Pairwise $F_{ST}$ values (below diagonal) and $F'_{ST}$ values (above diagonal) for 17 rhinoceros auklet breeding colonies based on nine microsatellite loci.**

| | | West | | | | | East | | | | | | | | | | | |
|---|---|---|---|---|---|---|---|---|---|---|---|---|---|---|---|---|---|---|
| West | | DAI | TAI | MAT | TEU | TOD | CH | MID | STL | LU | SGG | MO | TRI | PI | CL | DE | PR | AN |
| | DAI | ■ | 0.044 | 0.026 | 0.117 | 0.046 | 0.264 | 0.254 | 0.252 | 0.301 | 0.288 | 0.214 | 0.254 | 0.258 | 0.196 | 0.166 | 0.213 | 0.174 |
| | TAI | 0.013 | ■ | 0.075 | 0.228 | 0.107 | 0.202 | 0.215 | 0.254 | 0.254 | 0.284 | 0.188 | 0.251 | 0.221 | 0.138 | 0.152 | 0.192 | 0.166 |
| | MAT | 0.008 | 0.022 | ■ | 0.122 | 0.034 | 0.205 | 0.196 | 0.212 | 0.246 | 0.245 | 0.175 | 0.166 | 0.237 | 0.168 | 0.158 | 0.188 | 0.174 |
| | TEU | 0.036 | 0.071 | 0.038 | ■ | 0.034 | 0.302 | 0.272 | 0.284 | 0.307 | 0.275 | 0.269 | 0.223 | 0.248 | 0.293 | 0.305 | 0.283 | 0.244 |
| | TOD | 0.014 | 0.032 | 0.010 | 0.011 | ■ | 0.234 | 0.196 | 0.218 | 0.274 | 0.250 | 0.191 | 0.175 | 0.217 | 0.191 | 0.166 | 0.203 | 0.200 |
| East | CH | 0.086 | 0.067 | 0.067 | 0.105 | 0.078 | ■ | 0.009 | 0.010 | 0.008 | 0.013 | 0.045 | 0.037 | 0.000 | 0.034 | 0.070 | 0.019 | 0.051 |
| | MID | 0.083 | 0.071 | 0.064 | 0.093 | 0.065 | 0.003 | ■ | 0.037 | 0.005 | 0.009 | 0.062 | 0.037 | 0.019 | 0.053 | 0.080 | 0.076 | 0.093 |
| | STL | 0.070 | 0.075 | 0.064 | 0.090 | 0.066 | 0.003 | 0.012 | ■ | 0.045 | 0.041 | 0.052 | 0.029 | 0.027 | 0.031 | 0.054 | 0.062 | 0.039 |
| | LU | 0.104 | 0.090 | 0.086 | 0.112 | 0.098 | 0.003 | 0.002 | 0.016 | ■ | 0.000 | 0.098 | 0.064 | 0.012 | 0.066 | 0.124 | 0.097 | 0.136 |
| | SGG | 0.098 | 0.098 | 0.084 | 0.098 | 0.087 | 0.005 | 0.003 | 0.014 | 0.000 | ■ | 0.077 | 0.036 | 0.005 | 0.075 | 0.104 | 0.061 | 0.094 |
| | MO | 0.070 | 0.062 | 0.057 | 0.093 | 0.064 | 0.016 | 0.022 | 0.017 | 0.037 | 0.028 | ■ | 0.064 | 0.062 | 0.026 | 0.103 | 0.061 | 0.052 |
| | TRI | 0.081 | 0.081 | 0.055 | 0.076 | 0.058 | 0.013 | 0.013 | 0.010 | 0.024 | 0.013 | 0.023 | ■ | 0.033 | 0.081 | 0.098 | 0.099 | 0.103 |
| | PI | 0.084 | 0.073 | 0.078 | 0.085 | 0.072 | 0.000 | 0.007 | 0.009 | 0.004 | 0.002 | 0.022 | 0.012 | ■ | 0.043 | 0.103 | 0.069 | 0.074 |
| | CL | 0.056 | 0.039 | 0.048 | 0.089 | 0.055 | 0.011 | 0.017 | 0.009 | 0.022 | 0.025 | 0.008 | 0.025 | 0.014 | ■ | 0.030 | 0.033 | 0.061 |
| | DE | 0.048 | 0.044 | 0.047 | 0.095 | 0.049 | 0.023 | 0.026 | 0.016 | 0.043 | 0.035 | 0.034 | 0.032 | 0.034 | 0.009 | ■ | 0.032 | 0.024 |
| | PR | 0.066 | 0.060 | 0.059 | 0.094 | 0.064 | 0.007 | 0.026 | 0.020 | 0.035 | 0.022 | 0.021 | 0.034 | 0.024 | 0.010 | 0.010 | ■ | 0.037 |
| | AN | 0.052 | 0.049 | 0.052 | 0.077 | 0.060 | 0.017 | 0.031 | 0.012 | 0.048 | 0.032 | 0.017 | 0.033 | 0.024 | 0.018 | 0.007 | 0.012 | ■ |

One population and one locus were removed because of small sample size and missing data (Southeast Farallon Island and CMms26). Bold values indicate statistical significant at p < 0.05; with underlined values at p < 0.01 following corrections for multiple tests. Acronyms are DAI = Daikoku, TAI = Taijima, MAT = Matsumae, TEU = Teuri, TOD = Todojima (all in the western Pacific); and CH = Chowiet, MID = Middleton, STL = St. Lazaria, LU = Lucy, SGG = S'Gang Gwaay, MO = Moore, TRI = Triangle, PI = Pine, CL = Cleland, DE = Destruction, PR = Protection, AN = Año Nuevo (all in the eastern Pacific).

correction, 106 out of 136 tests were significant indicating a high level of genetic differentiation among all 17 remaining colonies.

The five western Pacific Ocean colonies were significantly differentiated at $p < 0.001$ from the 13 eastern Pacific Ocean colonies based on $F_{ST}$ values. The only exception was that Taijima in the western Pacific was significantly different at $p < 0.05$ from Destruction and Año Nuevo islands in the eastern Pacific, possibly due to the small sample size at Taijima ($n = 10$).

Among the western Pacific colonies, $F_{ST}$ significance varied between colonies. All five significant values included Teuri and Taijima, both of which were significantly different from three of the other four colonies (Table 3). Among the eastern Pacific breeding colonies, 41 of the 66 pairwise comparisons were significant (Table 3). Interestingly, the three Alaskan colonies (Chowiet, Middleton, St. Lazaria) accounted for most (16 of 25) of the non-significant values. Excluding those three sites in Alaska, 8 of the remaining 9 non-significant values were between nearby colonies, either in the central (Lucy-S'Gang Gwaay, Lucy-Pine, S'Gang Gwaay-Pine) or the southern (Cleland-Protection, Cleland-Destruction, Protection-Destruction, Protection-Año Nuevo, Destruction-Año Nuevo) regions of our study. The other non-significant pairwise comparison was between Cleland and Moore islands in British Columbia.

Visual inspection of the STRUCTURE and ΔK plots showed two distinct clusters, one including all five western Pacific Ocean colonies and one including all 13 eastern Pacific Ocean colonies. Most individuals had ancestry coefficient Q > 80% for one of the two clusters (Fig 2A). Comparison between the average lnPr(X|K) values at K = 2 (-10381) and K = 3 (-10352) indicated three genetically distinct populations. At K = 3, most individuals from the five western Pacific colonies had Q > 70% for the same cluster, but the eastern colonies split

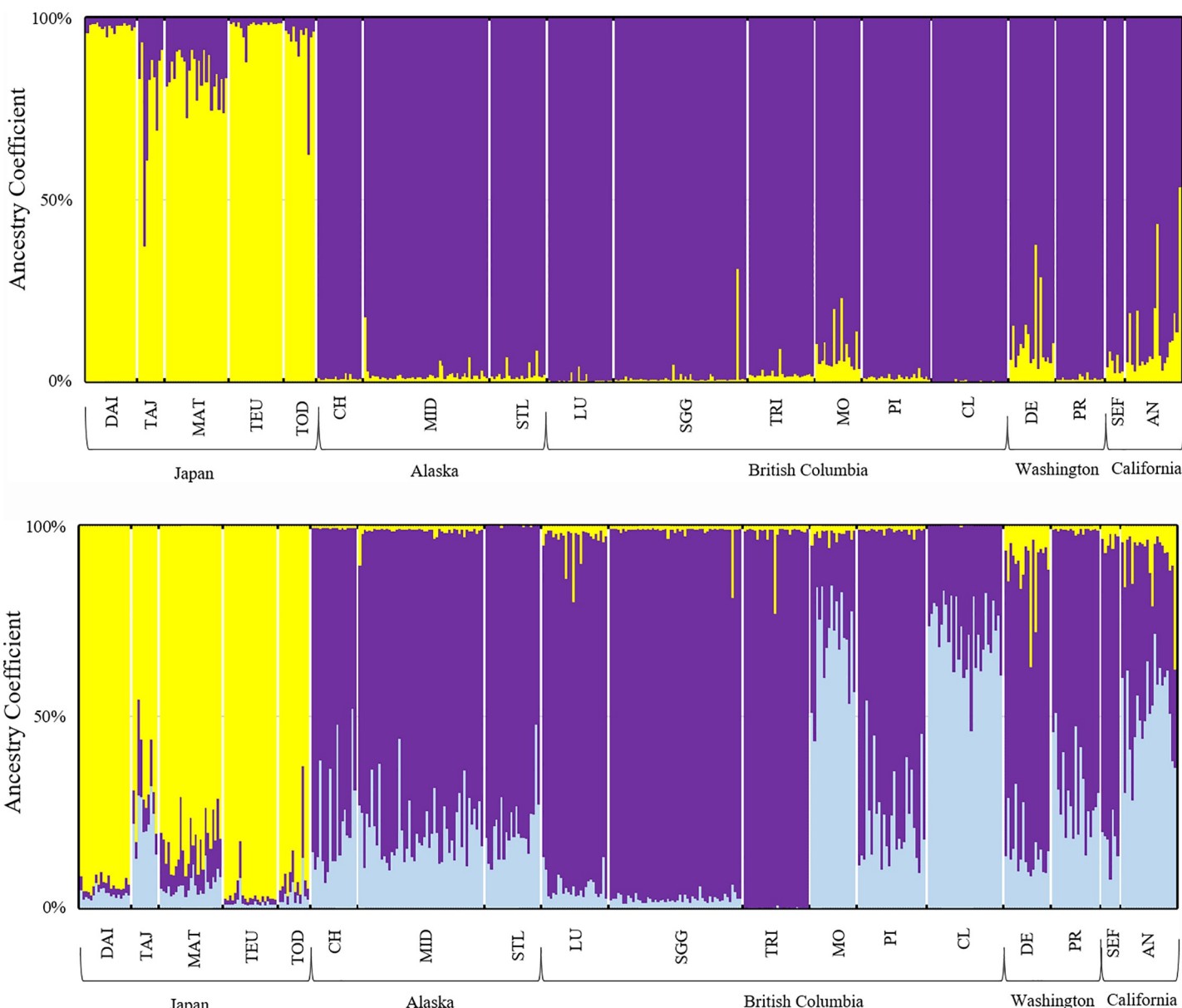

**Fig 2. Genetic groups for 18 North Pacific Ocean rhinoceros auklet colonies for ten microsatellite loci as inferred by STRUCTURE v2.3.** Histogram plot shows ancestry coefficient (Q) on the y-axis with individuals plotted on the x-axis. Breeding populations are listed from the western North Pacific (left) to eastern North Pacific (right). Genetic clusters at a) K = 2 and b) K = 3. (a) At K = 2 two genetic clusters occur between the western Pacific (Q > 70% yellow; Daikoku (DAI), Tajima (TAJ), Matsumae (MAT), Teuri (TEU), Todojima (TOD)) from the eastern Pacific (Q > 80% purple; Chowiet (CH), Middleton (MID), St. Lazaria (STL), Lucy (LU), S'Gang Gwaay (SGG), Triangle (TRI), Moore (MO), Pine (PI), Cleland (CL), Destruction (DE), Protection (PR), Southeast Farallon (SEF), Año Nuevo (AN)). (b) At K = 3 one cluster occurs for the western Pacific (Q > 70% yellow). For the eastern Pacific there are two clusters: Q > 60% for Chowiet, Middleton, St. Lazaria, Lucy, S'Gang Gwaay, Triangle, Pine, Destruction, Protection, Southeast Farallone (in purple); and Q > 60% for Moore and Cleland, with 60% of individuals from the Año Nuevo colony showing Q > 50% for cluster two (in light blue).

into two clusters: Q > 60% for one cluster consisting of Chowiet, Middleton, St. Lazaria, Lucy, S'Gang Gwaay, Triangle, Pine, Destruction, Protection, and Southeast Farallon; and Q > 60% for a second cluster consisting of Moore and Cleland, with 60% of individuals from the Año Nuevo colony showing Q > 50% for cluster two (Fig 2B).

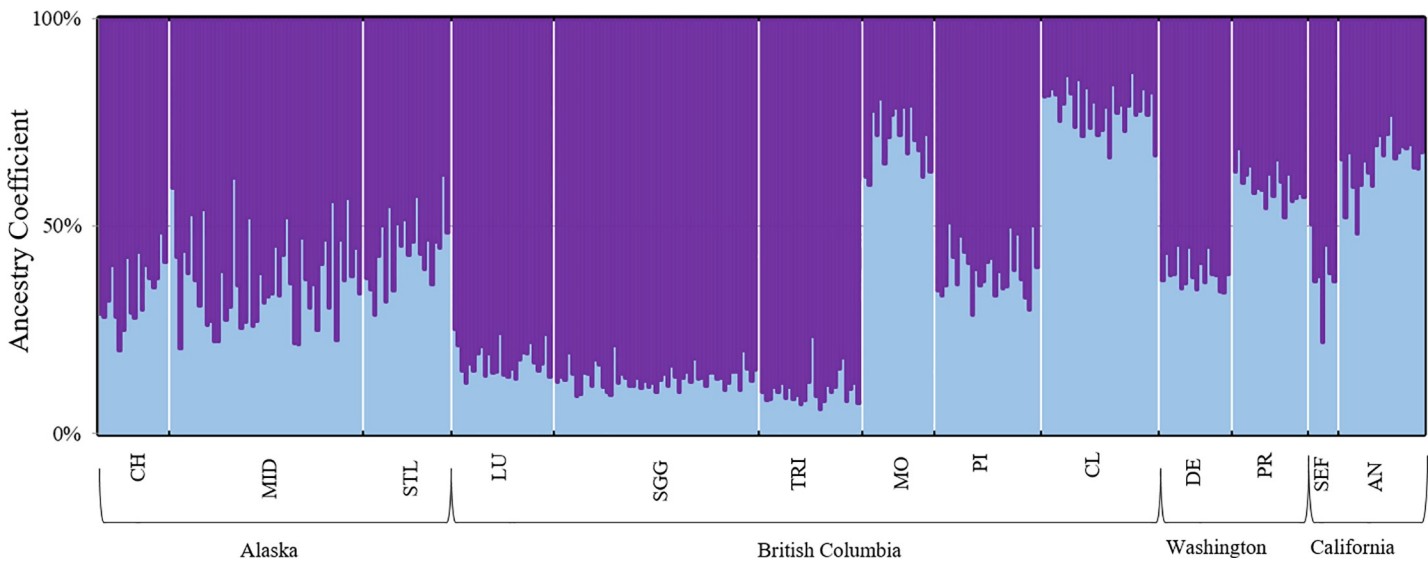

**Fig 3. Hierarchical analysis of 13 eastern Pacific rhinoceros auklet colonies for ten microsatellite loci as inferred by STRUCTURE v2.3.4.** Histogram plot shows ancestry coefficient (Q) on the y-axis with individuals plotted on the x-axis. Populations are listed as they move along the coastline from Alaska to California. Substructure was found for the 13 eastern breeding colonies, and supported by mean (ln Pr(X|K)) value and delta K (ΔK) at K = 2. Substructure includes two clusters at Q > 60%, one including Chowiet, Middleton, St. Lazaria, Lucy, S'Gang Gwaay, Triangle, Pine, Destruction, and Southeast Farallon (purple), and the second including Moore, Cleland, Protection, and Año Nuevo (light blue). No additional substructure was found among either the western or eastern breeding groups.

The hierarchical STRUCTURE analysis for the western Pacific colonies showed K = 1 (highest ln(Pr(X|K) = -2157). For the eastern Pacific colonies, the highest lnPr(X|K) value occurred at K = 2 (-8177). The eastern Pacific colonies split into two clusters at Q > 60%. The first cluster included Chowiet, Middleton, St. Lazaria, Lucy, S'Gang Gwaay, Triangle, Pine, Destruction, and Southeast Farallon, while the second cluster included Moore, Cleland, Protection and Año Nuevo (Fig 3).

The PCoA with ten loci and 17 breeding colonies showed clear separation between the western Pacific vs. eastern Pacific colonies. The first two axes explained 55.7% and 17.5% of the variation (third axis 8.4%; Fig 4) and the clustering is concordant with the STRUCTURE results. When the western Pacific colonies were examined alone, the first two axes explained 77.5% and 15.4% of the variation (third axis 6.8%). Three clusters were evident: Teuri with Todojima; Matsumae-Kojima with Daikoku, and Taijima. When the eastern Pacific breeding colonies were examined alone, the first two axes explained 47.3% and 16.5% (third axis 13.2%) of the variation. The majority of colonies in Alaska and British Columbia clustered together (Chowiet, Middleton, Lucy, S'Gang Gwaay, and Pine) with Triangle forming a separate cluster and the two WA colonies (Destruction and Protection) forming a third cluster. The remaining colonies of St. Lazaria, Moore, Cleland, and Año Nuevo showed some degree of separation from all of the other colonies.

## Geolocator tagging results

Of the total of 370 geolocator tags deployed on rhinoceros auklets, 150 tags were retrieved (Table 4). Due to logistical issues we were unable to retrieve tags on Moore Island in 2016. Retrieval rates varied widely by colony and year (x̄ = 40.3%, range = [0.0, 80.0]). We were able to obtain 142 lux files and 141 tracks following geolocation analyses of data from these devices, with some losses due to file corruptions or device failures.

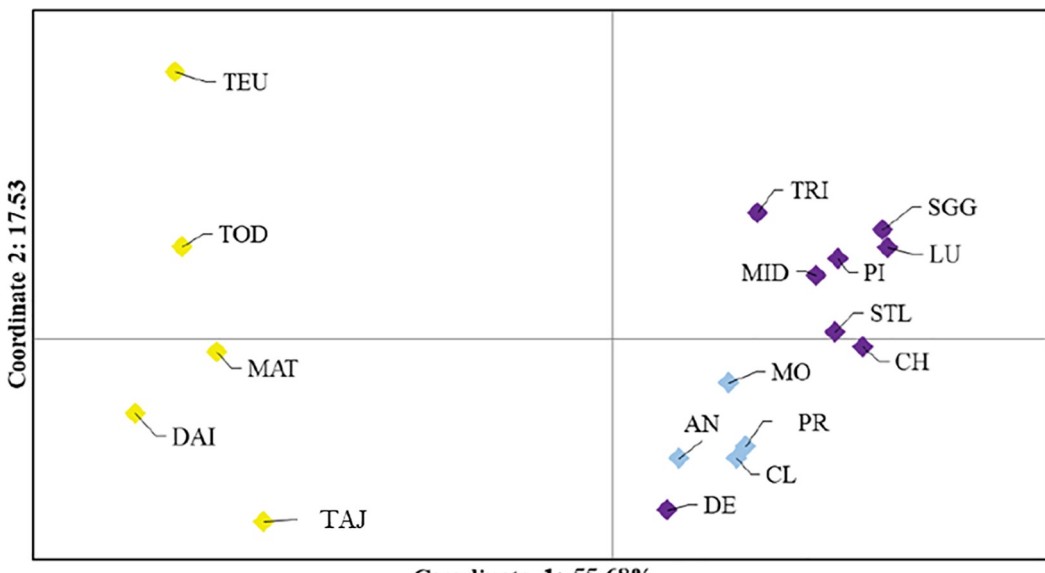

**Fig 4. PCoA analysis conducted in GenAlEx v6.5 based on pairwise F'<sub>ST</sub> values for the western and eastern North Pacific rhinoceros auklet breeding colonies.** Coordinates 1 and 2 explain 55.7% and 17.5% of the variation (not shown: coordinate 3 at 8.4%). The colours correspond to the three groups detected using STRUCTURE (Figs 2 and 3): western Pacific (yellow), the larger eastern Pacific group (purple) and smaller eastern Pacific group (blue).

Home range estimates from the geolocator tags varied widely by colony, but in general rhinoceros auklets remained in continental shelf waters during the winter months (Fig 5). The mean area of colony aggregates was 570,408 km$^2$ for 50 percent UD, and 2,216,996 km$^2$ for the 90 percent UDs (approximately 3.5x larger, a pattern that was consistent for individual-level UDs). For individual birds, the mean 50 percent UD had an area of 331,847 km$^2$, and 1,112,287 km$^2$ for the 90 percent UDs. In general, there was a latitudinal cline in home range areas, with markedly smaller home ranges for birds breeding at the two colonies in California, Southeast Farallon and Año Nuevo (Table 5, Fig 5).

Overall, the UDs from different colonies showed moderate spatial overlap during the winter months, with a mean UDOI of 0.45 across all pairwise values (Table 6). Spatial overlap during the winter months had a strong regional component, with the three Alaskan colonies all sharing much of the same area (UODIs ranging from 0.65 to 0.92). Similarly, the two colonies in California had an UDOI of 1.10, indicating they shared the same high use areas. Colonies from central and northern British Columbia (Lucy, S'Gang Gwaay, Triangle, Pine) all had high overlap with each other (UDOIs ranging from 0.83 to 1.34), and somewhat lower with Cleland Island along the west coast of Vancouver Island (UDOIs ranging from 0.54 to 0.72). Birds from the two colonies in Washington State, despite being in close proximity while breeding, appeared to have little overlap in winter (UDOI of 0.17). The wintering range of birds from Protection was similar to that of birds from British Columbia, whereas the two birds from Destruction were more strongly associated with the Pacific coast of the United States (Fig 5).

## Relationship between winter distribution and genetic structure

The genetic distance of rhinoceros auklets from the eastern Pacific colonies as measured by F'st was not correlated with the distance between the breeding colonies (IBD), with a Spearman correlation coefficient of 0.08 ($p$ = 0.290 from randomization; Fig 6A). In contrast, F'st

**Table 4. Sample sizes and retrieval rates of GLS devices deployed on rhinoceros auklets at breeding colonies along the Pacific Coast of North America, 2014–2015.**

| Colony | Year | Deployed | Retrieved | Retrieved (%) | Track files |
|---|---|---|---|---|---|
| Chowiet | 2014 | 14 | 5 | 35.7 | 4 |
| Middleton | 2014 | 20 | 16 | 75.0 | 16 |
| St. Lazaria | 2014 | 17 | 8 | 47.1 | 8 |
| Lucy | 2014 | 25 | 12 | 48.0 | 12 |
| Lucy | 2015 | 30 | 12 | 40.0 | 12 |
| Moore | 2015 | 5 | 0 | 0.0 | 0 |
| S'Gang Gwaay | 2014 | 30 | 11 | 36.7 | 10 |
| S'Gang Gwaay | 2015 | 31 | 14 | 45.2 | 14 |
| Triangle | 2014 | 30 | 13 | 43.3 | 12 |
| Triangle | 2015 | 31 | 16 | 51.6 | 16 |
| Pine | 2014 | 30 | 4 | 13.3 | 4 |
| Pine | 2015 | 22 | 5 | 22.7 | 4 |
| Cleland | 2014 | 20 | 2 | 10.0 | 2 |
| Protection | 2015 | 23 | 9 | 39.1 | 7 |
| Destruction | 2015 | 7 | 2 | 28.6 | 2 |
| Farallones | 2014 | 20 | 9 | 45.0 | 9 |
| Año Nuevo | 2014 | 15 | 12 | 80.0 | 9 |
| Total | | 370 | 150 | 40.3 | 141 |

was negatively correlated to the spatial overlap among colonies during the winter, with a Spearman's correlation coefficient of -0.22 ($p = 0.056$ from randomization; Fig 6B).

## Discussion

We combined light-level geolocator tracking with range-wide genetics analyses to directly test a key hypothesis often invoked to explain population genetic structuring in seabirds; specifically, that genetic differentiation is associated with spatial segregation in wintering areas [22, 28]. Our results support and build on a previous analysis [32] in showing that contemporary genetic structuring is extensive and complex in the rhinoceros auklet, a widely-distributed seabird of the temperate North Pacific Ocean. Genetic differentiation was strongest between the eastern vs. western Pacific populations, but structure also existed within both regional groups. Neither physical barriers to movement nor isolation by distance within the eastern group accounted for the pattern of structure. As predicted, geolocator tagging revealed a negative association between the extent of population differentiation between colony pairs (F'st) and the extent of overlap in non-breeding season distributions.

### Genetic structure and its relationship to winter distribution

The genetics component of our study, which spanned 18 breeding colonies and 424 individuals, further supports the major conclusions of the Abbott et al. study [32]. All of $F_{ST}$ and $F'_{ST}$ values, population-based PCoA, and individual-based STRUCTURE plots indicated that there is a high level of genetic differentiation between rhinoceros auklets breeding in the western vs. eastern Pacific Ocean.

Morphometric variation has been linked to genetic differentiation in marine vertebrates [71, 72], including seabirds [73, 74]. In the rhinoceros auklet, individuals breeding on colonies in the western Pacific are on average larger in linear dimensions and body mass than those in the eastern Pacific [75, 76]. On colonies involved in this study, the mean (SD) mass of 278 birds from four western Pacific colonies (all but Taijima) was 558 g (37.1), with a range of 460

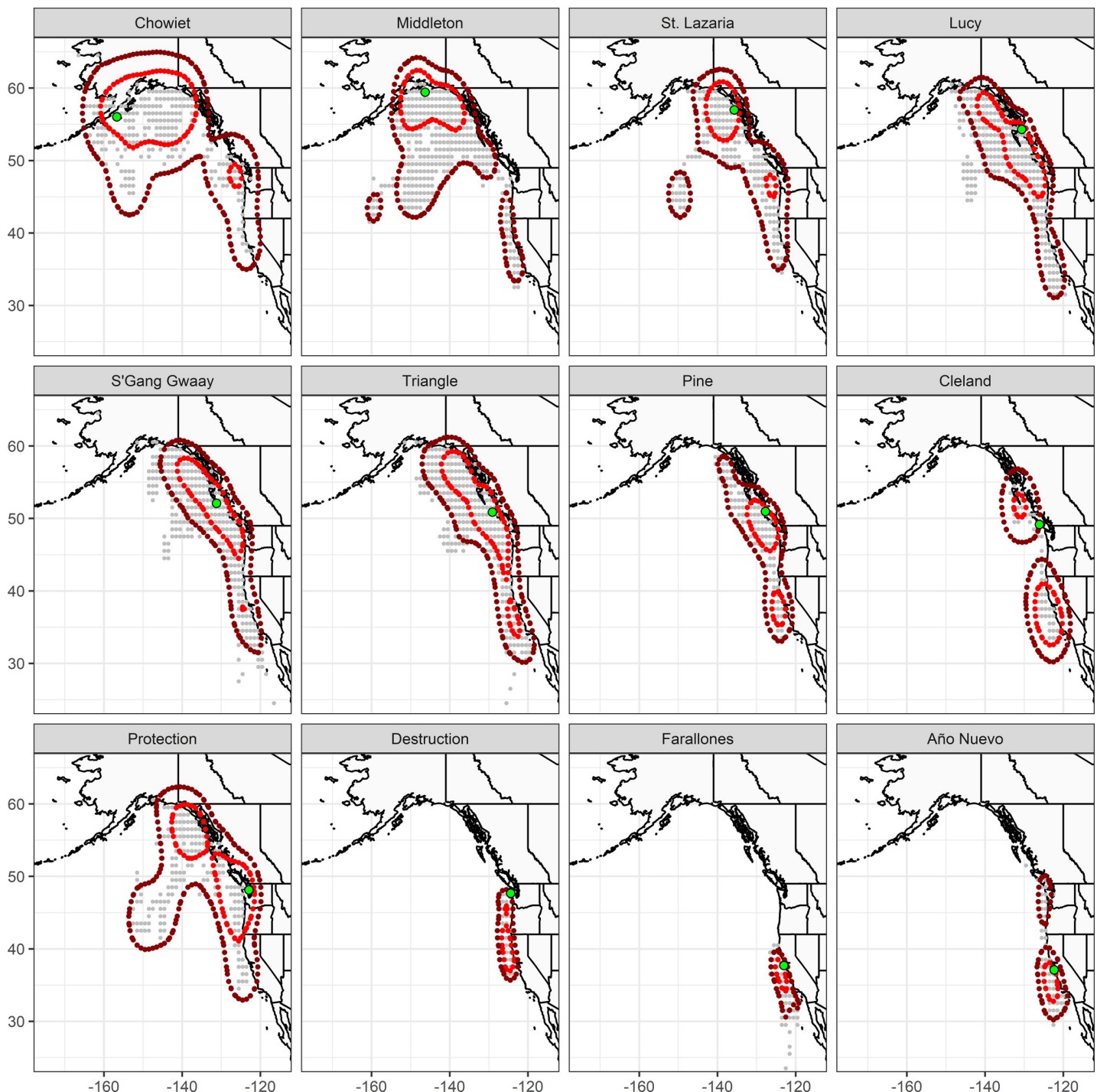

**Fig 5. Home ranges of rhinoceros auklets during the wintering period (Nov-Feb) from breeding colonies on the Pacific coast of North America, 2014 to 2016.** Light green dots indicate the locations of breeding colonies. Light gray marks are point locations on a 1˚×1˚ grid, as derived from TwilightFree geolocation analyses of light recorders deployed on all birds at each colony. Red polygons indicate the 50% Utilization Distribution ('core wintering area'), and dark red polygons indicate the 90% Utilization Distribution.

to 695 g; and 502 g (32.5) for 392 birds from all 13 eastern Pacific colonies, with a range of 425 to 620 g. There was little variation in mass within either region. No subspecies are currently

**Table 5. Home-range area estimates (km²) during wintering months (Nov-Feb) of rhinoceros auklets breeding at 12 colonies on the Pacific Coast of North America, 2014 to 2016, as calculated from the Utilization Distribution (UD) derived from geolocation of light loggers.**

| Colony | Colony (50p) | Colony (90p) | Individual Mean (50p) | Individual SD (50p) | Individual Mean (90p) | Individual SD (90p) |
|---|---|---|---|---|---|---|
| Chowiet | 1,415,726 | 5,120,609 | 866,352 | 875,273 | 2,875,308 | 2,643,150 |
| Middleton | 627,880 | 3,106,851 | 338,206 | 266,197 | 1,075,572 | 825,712 |
| St. Lazaria | 408,793 | 2,428,101 | 420,982 | 355,115 | 1,399,460 | 1,147,132 |
| Lucy | 650,667 | 2,311,713 | 502,806 | 486,398 | 1,610,695 | 1,550,292 |
| S'Gang Gwaay | 637,480 | 2,313,936 | 423,452 | 394,763 | 1,409,576 | 1,303,438 |
| Triangle | 837,116 | 2,687,246 | 473,992 | 378,839 | 1,496,879 | 1,150,994 |
| Pine | 404,399 | 1,505,507 | 136,205 | 125,384 | 468,058 | 377,293 |
| Cleland | 490,341 | 1,746,645 | 275,473 | 328,310 | 989,485 | 1,107,242 |
| Protection | 1,044,610 | 3,948,580 | 322,455 | 271,926 | 1,091,980 | 839,139 |
| Destruction | 121,136 | 429,640 | 87,846 | 7,867 | 350,103 | 65,891 |
| Farallones | 69,851 | 324,594 | 66,589 | 30,521 | 288,058 | 118,205 |
| Año Nuevo | 136,904 | 680,534 | 67,809 | 120,074 | 292,267 | 433,373 |

Colony refers to the UD derived from all birds at colony, and Individual refers to UDs of individual birds. Colonies are sorted from north to south (Fig 1).

recognized in the rhinoceros auklet [33], but given the marked genetic and morphological differences, the eastern and western Pacific Ocean populations might meet criteria for subspecific designation.

Abundant on both sides of the temperate North Pacific Ocean, the rhinoceros auklet is virtually absent as a breeder in the Aleutian Islands and Bering Sea, where two close relatives, the tufted puffin (*Fratercula cirrhata*) and horned puffin (*F. corniculata*), are abundant [77]. There are no obvious physical barriers to dispersal over that long gap in the distribution of the rhinoceros auklet. However, both a species distribution model (S1 Table, S1 Fig) and a resistance surface map (S2 Fig) developed in conjunction with this study [78] indicate that the deep ocean habitat along the northern continental shelf (Alaska to Asia) inhibits movement between populations in the eastern and western Pacific Ocean. In agreement with those analyses, no geolocator-tagged rhinoceros auklets crossed from the western to the eastern Pacific [35], nor

**Table 6. Spatial overlap between home ranges of during wintering months (Nov-Feb) of rhinoceros auklets breeding at 13 colonies on the Pacific Coast of North America, 2014 to 2016, as calculated from the Utilization Distribution (UD) derived from geolocation of light loggers.**

| | Middleton | St. Lazaria | Lucy | S'Gang Gwaay | Triangle | Pine | Cleland | Protection | Destruction | Farallones | Año Nuevo |
|---|---|---|---|---|---|---|---|---|---|---|---|
| Chowiet | 0.92 | 0.65 | 0.37 | 0.38 | 0.40 | 0.21 | 0.10 | 0.54 | 0.05 | 0.01 | 0.04 |
| Middleton | | 0.74 | 0.37 | 0.37 | 0.39 | 0.12 | 0.07 | 0.46 | 0.03 | 0.02 | 0.04 |
| St. Lazaria | | | 0.88 | 0.99 | 0.99 | 0.47 | 0.16 | 1.10 | 0.12 | 0.02 | 0.05 |
| Lucy | | | | 1.34 | 1.22 | 0.92 | 0.54 | 0.90 | 0.13 | 0.10 | 0.22 |
| S'Gang Gwaay | | | | | 1.26 | 0.92 | 0.56 | 0.96 | 0.18 | 0.10 | 0.21 |
| Triangle | | | | | | 0.83 | 0.62 | 0.88 | 0.17 | 0.13 | 0.27 |
| Pine | | | | | | | 0.72 | 0.70 | 0.34 | 0.15 | 0.33 |
| Cleland | | | | | | | | 0.31 | 0.34 | 0.46 | 0.78 |
| Protection | | | | | | | | | 0.17 | 0.03 | 0.12 |
| Destruction | | | | | | | | | | 0.15 | 0.24 |
| Farallones | | | | | | | | | | | 1.10 |

Overlap measures are calculated as a UD overlap index (UDOI) between each pairwise combination of colony UDs (Fieberg and Kochanny 2005). The UDOI is 1.0 when both UDs are uniformly distributed and share 100% overlap, 0 when UDs have no overlap, and > 1.0 when the two UDs are non-uniformly distributed and share high overlap. Colonies are sorted from north to south (Fig 1).

## a. Distance between breeding colonies

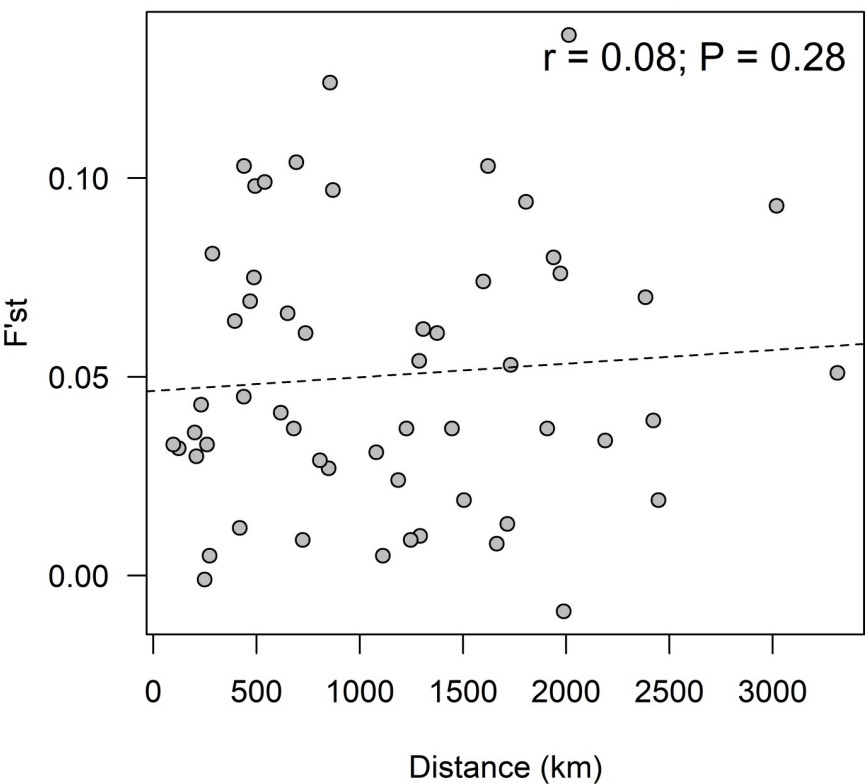

## b. Spatial Overlap during winter

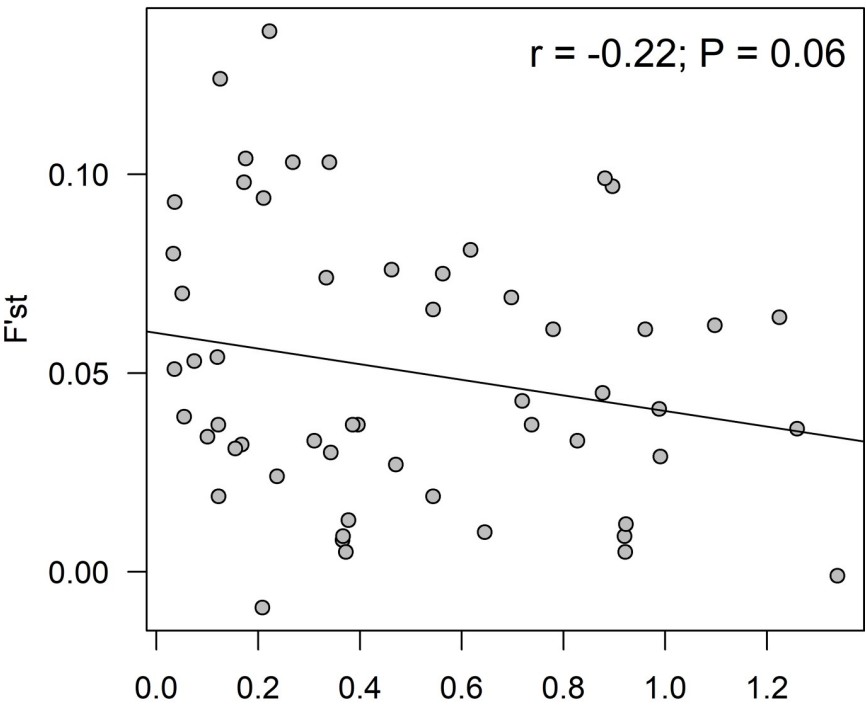

**Fig 6.** Correlations between genetic distance, as measured by standardized measure of genetic differentiation (F'st), and (a) distance between breeding colonies (IBD), or (b) spatial overlap during the non-breeding season (Nov-Feb). Spatial overlap was measured by the Utilization Distribution overlap index (UDOI) between each pairwise combination of colonies of rhinoceros auklets at breeding colonies on the Pacific Coast of North America, 2014 to 2016.

from the eastern to the western Pacific (this study), in the non-breeding season. Cryptic barriers to dispersal limit gene flow in other seabirds as well [79, 80]. However, in another auk species, the ancient murrelet (*Synthliboramphus antiquus*), many individuals do cross from the eastern to the western North Pacific over their annual life cycle [81]. And as predicted from the hypothesis that use of common wintering areas is one factor that links with gene flow among populations, no genetic differentiation was found between ancient murrelets breeding in the eastern vs. western Pacific based on mitochondrial DNA [82].

There is historical context to consider as well. In other marine vertebrates, ice cover during the late Pleistocene appears to have promoted an east-west genetic split across the North Pacific Ocean [83–85]. For the rhinoceros auklet, it is plausible that extensive ice cover over the North Pacific during the last glacial maximum ~19–14.6 kya [39] forced a southward range shift that isolated populations into eastern and western refugia, thereby leading to genetic divergence [32].

While genetic differentiation in rhinoceros auklets was strongest between the eastern and western North Pacific Ocean, there was a striking amount of structure within both regional groups as well. Based on $F'_{ST}$ values, five of 10 comparisons between pairs of colonies were significant in the western Pacific Ocean, as were 41 of 66 pairwise comparisons in the eastern Pacific Ocean. That smaller-scale differentiation exists in the absence of obvious physical barriers to dispersal, and could not be attributed to isolation by distance. Such a complex pattern of genetic differentiation is unusual among seabirds [47], and there are undoubtedly many factors involved [27]. Here, we will consider the role of three primary factors that could be associated with genetic structuring in rhinoceros auklets: (1) philopatry; (2) a mainly coastal (neritic) foraging habitat; and (3) the extent of overlap in wintering areas.

**Philopatry.** Philopatry to breeding sites is a common behavioural trait in marine vertebrates [86, 87], including colonial seabirds [88]. Like other colonial auks, the rhinoceros auklet appears to be highly philopatric [34]. Between 1984 and 2019, over 4000 individuals, including both adults and nestlings, were banded on the colonies in British Columbia included in this study. Many hundreds have been re-encountered, but only on the colonies where they were banded [89, 90].

Nonetheless, some movement among colonies does occur. A number of rhinoceros auklet breeding colonies were newly established, or re-established, following extirpation due to human influence in Oregon and California from the 1960s to the 1990s [91–93], including the Farallon Islands [94] and Año Nuevo Island [95]. There were clues from Fst values about the scale and nature of movement in rhinoceros auklets. Only 25 of 66 pairwise comparisons among the eastern Pacific colonies were non-significant, and 16 of those 25 involved the three breeding colonies in Alaska. Of the nine remaining non-significant comparisons, eight involved nearby pairs of sites: five at the southern half of the range (California to southern British Columbia, all situated within the California Current Marine System), and three further north (central and northern British Columbia). Those observations imply, first, that individuals breeding in Alaska are more likely to move away from their natal colonies than individuals breeding elsewhere, a notion consistent with the generally large wintering ranges of Alaskan birds; and second, that movements of individuals on breeding colonies south of Alaska, when they occur, tend to be regional.

**Neritic foraging habit.** The majority of rhinoceros auklets breeding in the eastern North Pacific Ocean inhabit colonies in British Columbia, and the largest colonies are located along the mainland coast [44]. That distribution presumably reflects the importance of Pacific sand lance (*Ammodytes personatus*) to these seabirds [96–98]: the availability of that specific prey species is closely tied to the annual productivity of rhinoceros auklet colonies in British Columbia [99, 100]. The Pacific sand lance, like other species of sand lances, inhabits shallow, coastal environments [101]. The fact that deep ocean habitat across the North Pacific Ocean appears to act as a barrier to movement between eastern and western rhinoceros auklet populations provides further evidence of the neritic nature of the species.

There is evidence in several seabird taxa, including albatrosses [102], boobies [103] and penguins [79], and in some marine mammals [104], that species that forage predominantly in coastal, neritic environments tend to exhibit more extensive genetic structuring than species that forage in open-ocean, pelagic environments. That appears to be true among the auks of the North Pacific Ocean as well. Genetic structure is prominent in three coastal species, the pigeon guillemot (*Cepphus columba*) [105], marbled murrelet (*Brachyramphus marmoratus*) [106] and Kittlitz's murrelet (*B. brevirostris*) [107]. But structure is less prominent in the more oceanic common murre (*Uria aalge*) [108], thick-billed murre (*Uria lomvia*) [109], ancient murrelet [82], Cassin's auklet (*Ptychoramphus aleuticus*) [50], and crested auklet [110]. A potentially confounding factor is that all of the species in the latter group breed in large colonies, whereas all of the species in the former group breed solitarily or in small aggregations [33]. But the rhinoceros auklet nests in large colonies, is mainly neritic, and exhibits extensive genetic structuring. Those same traits are also characteristic of the whiskered auklet [111], a species that forages in tide-rips close to shore [111], and a species in which birds outfitted with geolocator tags attended the breeding colony year-round [112]. Neritic seabirds might tend to structure extensively because they conform to a one-dimensional, stepping-stone model of dispersal [105], and/or because they are unlikely to come into contact at sea with individuals from populations other than their own [79].

**Overlap in wintering areas.** Two lines of evidence in our results supported the hypothesis of a link between segregation in wintering areas and population genetic structuring. First, none of the tagged rhinoceros auklets breeding on one side of the North Pacific Ocean crossed to the other side [35, this study], and there was strong differentiation between populations in the eastern vs. the western Pacific. And second, in comparisons between pairs of colonies in the eastern Pacific, there was a negative correlation between the standardized measure of genetic differentiation (F'st) and the extent of spatial overlap in wintering areas. The wintering range of rhinoceros auklets breeding on colonies in North America encompasses nearly all continental shelf waters of the eastern North Pacific Ocean. On average, spatial overlap in winter was moderate among all colonies (UDOI = 0.45), and there was some regional structure, especially for the two colonies in California (UDOI values > 1) that had similar high use areas in winter, not just broad overlap.

It is most often suggested that segregation in wintering areas is associated with genetic differentiation because segregation reduces the chance that an individual from one population will encounter foreign breeding colonies and/or individuals from other populations [22, 27]. For rhinoceros auklets, the strong differentiation between western vs. eastern Pacific populations appears to be caused in large part by a more-or-less impermeable barrier to dispersal in the form of deep ocean habitat in the Aleutian Islands and Bering Sea–perhaps equivalent in effect to differentiation in other seabirds that inhabit separate ocean basins and in which genetic exchange is extremely rare [113] or non-existent [108]. Historical isolation during the late Pleistocene also could have led to differentiation between the eastern and western Pacific Ocean populations. Not surprisingly, the differentiation within the two populations of

rhinoceros auklet was more subtle, and in the eastern population at least, linked to the *degree* of segregation in wintering areas. That, we propose, is consistent with the idea that the probability of encountering foreign colonies and/or individuals at sea is a mechanism linking distribution to genetic structure.

Other mechanisms are also plausible. In Cook's petrel (*Pterodroma cookii*), genetic differentiation was promoted by population-level differences in the timing of breeding, driven by differences in migration timing associated with habitat specialization in wintering areas [31]. But allochrony seems an unlikely primary mechanism for rhinoceros auklets. On colonies in the eastern Pacific, timing of breeding becomes later with increasing latitude, e.g., the date of laying of the first egg in the year ranged from 8 April to 11 May in 12 years on Año Nuevo Island, CA (Oikonos, unpublished data); 30 April to 7 May in 5 colony-years on Protection and Smith islands, WA [114]; 22 April to 4 May in 5 years on Triangle Island, BC [115]; and 4 to 17 May in 3 years on the Semidi Islands, AK [116]. But while there is a latitudinal trend in laying dates, there was no strict latitudinal pattern of population differentiation.

Although the Friesen, Burg, & McCoy [22] hypothesis has not been directly tested in other auks, separate investigations of winter distribution and genetic structure provide further support, and suggest areas for future research. First, the ancient murrelet is panmictic across the North Pacific Ocean [82], and geolocator tagging revealed that some individuals crossed from the eastern into the western Pacific in the winter [81]. Second, population genetic structure exists in the whiskered auklet [111], and geolocator tagging showed that individuals in this species overwinter close to breeding colonies [112]. Third, the winter distribution of most Cassin's auklets breeding on two very large colonies in British Columbia (Triangle, Frederick islands) overlapped with that of individuals from Southeast Farallon Island in California [117, 118], and the two groups are genetically homogeneous [50]. And fourth, analysis of highly differentiated loci in thick-billed murres from five colonies in Canada's eastern Arctic showed that individuals from the three colonies showing highest overlap in wintering distributions also were those that were the most genetically similar [119].

But there is also contradictory evidence among the auks. Genetic structure was more extensive in Atlantic Ocean populations of common murres [108] than in thick-billed murres [109], whereas geolocator tagging in the western Atlantic found that population segregation in winter was stricter in thick-billed than in common murres [120]. That apparent contradiction could signal the existence of real variation among auks, even between close relatives, or differences in the scale of the genetics vs. the tagging projects. Finally, a range-wide geolocator tagging study of Atlantic puffins (*Fratercula arctica*) [121] revealed a complex pattern of overlap in wintering areas, whereas a much smaller-scale study found little genetic differentiation based on allozyme patterns [122]. An investigation of population genetic structure in the Atlantic puffin to match the scale of the geolocator tagging could provide much insight.

## Conservation implications

Large-scale tagging programs are providing valuable new information on the year-round habitat use of seabirds [121], with direct application to marine conservation [123]. Seabirds are particularly wide-ranging organisms of high conservation concern, they face a wide array of anthropogenic threats both on the land and at sea, and many species are experiencing population declines [124]. Geolocator tagging revealed that the rhinoceros auklet is very widely distributed throughout the year in continental shelf waters of the temperate North Pacific Ocean. Within this realm, rhinoceros auklets are vulnerable to a wide variety of anthropogenic stressors including oiling at sea [125, 126], bycatch in fishing gear [127–129], and chemical contamination of food webs [130–133]. They are also affected by oceanographic change [100, 134].

Widespread and vulnerable, the rhinoceros auklet also displays an extensive and complex pattern of population genetic structure. The maintenance of genetic diversity is critical to ensuring that species remain resilient to natural and anthropogenic stressors over the long term [135, 136], and is a particularly challenging component of biodiversity conservation [137, 138]. Thus, many issues uncovered here, for one North Pacific species, indicate the scope and complexity of the conservation challenges posed by seabirds.

## Supporting information

**S1 Table. Environmental layers used to develop the Species distribution model.** Contributions to the model are determined using a heuristic approach that depends on the path of the Maxent code. Permutation importance is determined by values randomly permutated along training points and measurements for the decrease in training AUC. Variables with a higher influence have a larger percent value.
(DOCX)

**S1 Fig. Species distribution model for rhinoceros auklet created from GBIF breeding season occurrences (May-July).** Map was produced using the SDM toolbox (Brown, 2014; Brown et al., 2017), Maxent (Phillips et al., 2006; Phillips & Dudík, 2008), and ArcMap. The final map was visualised using ArcMap 10.2 (ESRI®) in the Azimuthal Equidistant (180° meridian) projection. Most suitable habitat for is shown in cool (blue) unsuitable habitat in warm (orange to red).
(TIF)

**S2 Fig. Resistance surfaces for 18 rhinoceros auklet breeding colonies located throughout the North Pacific Ocean.** Areas of low resistance are in blue with gradients of orange to red indicating area of higher resistance. Breeding colonies denoted by the pink circled star. Resistance surfaces were developed using friction surfaces from an inverted SDM for rhinoceros auklet and least-cost corridors function within the SDM toolbox (Brown, 2014; Brown et al., 2017). The final map was visualised using ArcMap 10.2 (ESRI®) in the Azimuthal Equidistant (180° meridian) projection.
(TIF)

## Acknowledgments

Many people assisted us with sample collection and geolocator deployments, and we thank them all. For supporting our work on colonies in British Columbia we thank the Ahousaht, Gitga'at, Haida, Kitasoo/Xai'xais, Metlakatla, Quatsino, and Tlatlasikwala First Nations. Safe transport to field sites in BC was provided by the Canadian Coast Guard, Parks Canada, and West Coast Helicopters. For permits to conduct the research, we thank the Bird Banding Laboratory of the United States Geological Survey, the United States Fish and Wildlife Service, the Alaska Maritime National Wildlife Refuge, British Columbia Parks, and Environment and Climate Change Canada. Finally, we thank J. Bossart, A. Tigano, and an anonymous reviewer for their constructive feedback on the initial draft of the paper.

## Author Contributions

**Conceptualization:** J. Mark Hipfner, Glenn T. Crossin.

**Formal analysis:** Marie M. Prill, Aidan D. Bindoff, Mark C. Drever, Theresa M. Burg.

**Funding acquisition:** J. Mark Hipfner, Katharine R. Studholme, Jessie N. Beck, Russell W. Bradley, Ryan D. Carle, Glenn T. Crossin, Theresa M. Burg.

**Investigation:** J. Mark Hipfner, Marie M. Prill, Katharine R. Studholme, Alice D. Domalik, Strahan Tucker, Catherine Jardine, Mark Maftei, Kenneth G. Wright, Russell W. Bradley, Ryan D. Carle, Thomas P. Good, Scott A. Hatch, Peter J. Hodum, Motohiro Ito, Scott F. Pearson, Nora A. Rojek, Leslie Slater, Yutaka Watanuki, Alexis P. Will, Glenn T. Crossin, Mark C. Drever.

**Methodology:** Marie M. Prill, Jessie N. Beck.

**Project administration:** J. Mark Hipfner, Katharine R. Studholme.

**Supervision:** J. Mark Hipfner, Glenn T. Crossin, Theresa M. Burg.

**Writing – original draft:** J. Mark Hipfner, Marie M. Prill, Theresa M. Burg.

**Writing – review & editing:** J. Mark Hipfner, Katharine R. Studholme, Alice D. Domalik, Strahan Tucker, Catherine Jardine, Mark Maftei, Kenneth G. Wright, Jessie N. Beck, Russell W. Bradley, Ryan D. Carle, Thomas P. Good, Scott A. Hatch, Peter J. Hodum, Motohiro Ito, Scott F. Pearson, Nora A. Rojek, Leslie Slater, Yutaka Watanuki, Alexis P. Will, Aidan D. Bindoff, Glenn T. Crossin, Mark C. Drever, Theresa M. Burg.

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
