## [Decision Letter · Decision Letter 0]

21 May 2020

PONE-D-20-09231

Geolocator tagging links distributions in the non-breeding season to population genetic structure in a sentinel North Pacific seabird

PLOS ONE

Dear Dr. Hipfner,

Thank you for submitting your manuscript to PLOS ONE. After careful consideration, we feel that it has merit but does not fully meet PLOS ONE’s publication criteria as it currently stands. Therefore, we invite you to submit a revised version of the manuscript that addresses the points raised during the review process.

We would appreciate receiving your revised manuscript by Jul 05 2020 11:59PM. To enhance the reproducibility of your results, we recommend that if applicable you deposit your laboratory protocols in protocols.io, where a protocol can be assigned its own identifier (DOI) such that it can be cited independently in the future. For instructions see: http://journals.plos.org/plosone/s/submission-guidelines#loc-laboratory-protocols

We look forward to receiving your revised manuscript.

Kind regards,

Janice L. Bossart

Academic Editor

PLOS ONE

Journal Requirements:

2. In your Methods section, please state the volume of the blood samples collected for use in your study.

3. In your Methods section, please provide additional information regarding the permits you obtained for the work. Please ensure you have included the full name of the authority that approved the field sites access and, if no permits were required, a brief statement explaining why.

Additional Editor Comments (if provided):

Please pay close attention to and address concerns raised by both reviewers. Both make legit points. Major concerns that need to be addressed are:

1) General issues regarding IBD, e.g. whether appropriate or not across the entire range, and needing to be introduced and conceptually developed within the Introduction.

2) Correlation vs. causation.

3) Clarity regarding breeding vs. non-breeding/winter areas, timing of such, etc... e.g. as identified by Reviewer 2. To the extent that genetic structure is a consequence of gene flow, which of course is tied directly to mating, there needs to be better description of these colonies and where/when breeding takes place versus where/when they are relative to this when not breeding. Like Reviewer 2, I was confused. The reader is provided insufficient information to understand the situation.

In addition to the other issues the two reviewers identified, below is a list of minor problems I found.

Throughout, replace the apostrophe in F'ST with a straight apostrophe (e.g. using insert symbol in Word).

Figure 2a & 2b. Double check the ordering/naming of the colonies. To the extent, n is larger for MAT than TEU, these two, at least, apparently switched order between a & b. I also wondered about DE & PR. Be 100% sure that all names are associated with the correct genetic info in the STRUCTURE diagram.

Also, Figure 2a & 2b. Both should look identical structurally given they are essentially part of the same figure: 2a currently has more space and an additional solid black line on the left. The region names and brackets also need to better match the associated breeding colonies. Currently, TOD looks to be part of Alaska, especially in 2b. The brackets and region names in general appear to be less carefully aligned/applied in 2b versus 2a. There's also no reason the bracket for Japan should extend well beyond the colony info. Making things exact is easy using PowerPoint by inserting figures, then adding text/brackets and aligning using grid lines. Seems to me 2a and b could have easily been pre-assembled into a single jpeg.

Seems to me given their geographic location and their genetic similarity, SGG and TRI should be ordered together in all STRUCTURE figures/diagrams.

In general, bracket lines on all STRUCTURE figures should line up with the vertical bracket lines in the STRUCTURE diagram.

Figure 4. Relevance of colors and legend. Change label for Protection to PR.

Figure 6. Correlation coefficients are generally designated using 'r' not 'p'. Until I read the text, I thought the figure included p-values of non significance. For clarity, replace 'p' with 'r'.

Line 393 and 394. Replace Table 4 with Table 3.

Line 419. Replace 'Moore, Cleland' with 'Moore and Cleland'

Line 420. Identify which cluster

Line 436-440. Given that you're dealing with Q>60% throughout, use it once only... or at most twice. E.g. The eastern Pacific colonies split into two clusters at Q>60%. Cluster one included... etc. Cluster two included... In cluster one, delete 'Protection'. In both cases, precede the last colony in the list by 'and'. Why were SEF and MID singled out given all member of cluster 1 and cluster 2 were associated with the respective clusters at Q>60%?

Line 447. Delete Protection. The text says 60%. Which is it 50 or 60%?

Line 448. Delete 'and' before Protection. Replace '&' with 'and' before Ano Nuevo. The text says 60%. Which is it 50 or 60%?

Line 520. Figure 6b says -0.21. Which is it?

Lines 641-643. Given the amount of spread, i.e. low correlation, and fact the correlation was only significant at 0.05, I think this is a bit overstated.

Reviewers' comments:

Reviewer's Responses to Questions

**Comments to the Author**

1. Is the manuscript technically sound, and do the data support the conclusions?

Reviewer #1: Yes

Reviewer #2: Yes

2. Has the statistical analysis been performed appropriately and rigorously? 

Reviewer #1: Yes

Reviewer #2: Yes

3. Have the authors made all data underlying the findings in their manuscript fully available?

Reviewer #1: Yes

Reviewer #2: Yes

4. Is the manuscript presented in an intelligible fashion and written in standard English?

Reviewer #1: Yes

Reviewer #2: Yes

5. Review Comments to the Author

Reviewer #1: This paper contributes to our understanding of the factors promoting population differentiation in seabirds and other species. I really enjoyed reading this paper. It's written very well and and it is clear and easy to follow. The science is sound and the methods are appropriate to test the hypotheses presented in the introduction. The conclusions are supported by the data and are well presented and discussed in a comparative framework as well. I have only very minor edits to suggest.

371-372: negative FST values are an artifact of some programs and should be replaced by zeros.

389-390: A significant Mantel’s test indicates correlation between genetic and physical distance but this doesn’t translate necessarily into isolation by distance. The fact that you get a significant Mantel’s test when you include populations from both sides of the Pacific but not when you analyze eastern Pacific colonies alone (lines 516-518) indicate that in the former the pattern is driven by strong pop structure between eastern and western Pacific colonies rather than by IBD. See Meirmans, P.G. 2012. The trouble with isolation by distance. Molecular Ecology 21: 2839-2846.

516-518: I would say that all results from Mantel’s should be presented together but testing for IBD between eastern and western Pacific colonies may be inappropriate (see above).

547-548: See comments above. This is structure between two differentiated groups rather than IBD.

Line 682: Analysis of highly differentiated loci in thick-billed murres (Tigano et al. 2017) has revealed a similar pattern, although not quantitively. The three Atlantic colonies that were more genetically similar at these outlier loci were also the colonies showing the highest overlap in wintering distributions. However, this pattern could have also been explained by colonization history following the retreat of the Pleistocene ice sheets or selection during the non-breeding time.

Figure 1. Please show all colonies included in the study and highlight those that have also GLS data

Figure 2-4 Structure plot image quality is low and pop labels are hard to read. Make sure that images are high quality.

Fig. 4 What do colors represent? Add legend.

Fig. 6 Add p-values on graphs and/or figure caption.

Reviewer #2: This paper makes an important contribution to our understanding of population structure, distribution, and space use of the Rhinoceros Auklet. The study is particularly valuable because it aggregates data from breeding colonies across nearly the entire distribution of the species, which will be critical baseline information for management and conservation of the species. The data are robust and well analyzed. Overall this is a well researched and well written paper!

My only real issue with the paper is that the abstract, intro, and discussion use language that makes it appear the authors found a causal relationship when they only tested a correlation. The study is framed as a test of the hypothesis that segregation in the wintering distributions drives genetic differentiation among breeding colonies. However, the paper does not test a causal mechanisms by which nonbreeding segregation would drive genetic differentiation, and as a paper the authors cite (Rayner et al 2011) points out, “Spatial segregation during the non-breeding season appears to provide an intrinsic barrier to gene flow among seabird populations that otherwise occupy nearby or overlapping regions during breeding, but how this is achieved remains unclear.” What the study does do is test if there is a correlation between genetic differentiation and degree of difference in space use during the nonbreeding season. I think the language can easily be revised to clarify that this distinction by removing words like “promotes” and “drives” that imply causation and replacing them with words like “correlates with” that make it clear what the authors are testing.

The authors only set up one hypothesis test in the intro (winter segregation is correlated with genetic structure), but they test an additional hypothesis in the paper (distance between breeding colonies is correlated with genetic structure). I assume the authors test this to evaluate if average dispersal difference accounts for genetic structure, assuming all populations have equal mean dispersal differences (which is not a given). Since the authors present results of both tests and use them to say that overwintering overlap better accounts for genetic difference than mean isolating distance, both hypothesis tests should be set up in the intro. The authors also discuss two additional hypotheses in the discussion: genetic structure correlates with egg lay date or morphological differences like body size or mass (potentially indicating differences in foraging niche). These are also testable hypotheses, and it appears the authors have the data to test them instead of just discussing them. I think this could potentially strengthen the paper, though it’s not essential.

A limitation of the study is that the authors did not consider temporal segregation, only spatial segregation. Because of that, it is unclear to what degree these breeding colonies are potentially in contact during the nonbreeding season. The geolocator study cited from Japan (Takahashi et al. 2015) did find that space use varied with month, so I’m curious why the authors did not consider or account for that. I think the authors should address this in the discussion at a minimum.

Authors use inconsistent terminology for geolocators. I found the following terms used in the ms: geolocator, geologger, logger, and GLS device. Bird carrying geolocators were also referred to as loggered and geologgered. Best to pick one and use it consistently to avoid confusion.

325: When do the colonies initiate and end breeding? How does that relate to the “winter” dates selected?

365, comma missing

631-633: I found this sentence confusing. I recommend “in which the population showed the unusual behavior of remaining in close proximity to a breeding colony year-round and foraging…”

640-643: change word from “promotes” as you don’t have evidence of a mechanistic relationship.

658: not “driven,” correlated with. Again, no causation has been shown.

669-670: These are alternative hypotheses that are testable.

Table 4: I think there’s an error in Middleton, which shows more track files recovered than geolocators recovered.

Fig 1: As this is the only full study area map, the Western Pacific sites should be included and differentiated with a different color or icon.

Fig 4: Unclear what the colors represent. Please include a color legend. It would be logical for colors to be grouped by the five regions identified in figures 2 and 3, but there are more than five colors presented here. Please clarify.

Figure 5: The resolution of the file attached for figure 5 is low, so it’s hard to figure out what the gray lines are. The caption says they represent the geolocator tracks, but the look like dashed, horizontal lines, whereas geolocator tracks are typically presented as a single connected line. The gray lines here look more like shading, but it’s unclear how that shading was generated. Does this incorporate some metric of uncertainty around each point in the track or aggregate all tracks in some way? Please clarify.

6. PLOS authors have the option to publish the peer review history of their article (what does this mean?). If published, this will include your full peer review and any attached files.

Reviewer #1: Yes: Anna Tigano

Reviewer #2: No

---

## [Author Response · Author response to Decision Letter 0]

1 Sep 2020

Journal Requirements:

We have carefully reviewed the document to ensure complaince with PLoS ONE format.

2. In your Methods section, please state the volume of the blood samples collected for use in your study.

We added the following information: Blood samples (1.0 ml) and/or feather samples (5-10, from the breast) were collected from seven to 80 individuals on each of the 18 colonies, for a total of 704 individuals [line 225].

3. In your Methods section, please provide additional information regarding the permits you obtained for the work. Please ensure you have included the full name of the authority that approved the field sites access and, if no permits were required, a brief statement explaining why.

We have provided all information on permits to access field sites and conduct the research [lines 211-219].

We have deposited the microsatellite data in Dryad and the GLS tagging data in Movebank.

Comments from the Editor

1) General issues regarding IBD, e.g. whether appropriate or not across the entire range, and needing to be introduced and conceptually developed within the Introduction.

We agree with Reviewer #1 that our test for IBD across the full range of the rhinoceros auklet was not appropriate, and have removed this. We have retained the test for IBD among the eatsern Pacific Ocean colonies (finding no support), and have briefly introduced the topic in the Introduction using the helpful phrase from Reviewer #2 [lines 165-167]. 

2) Correlation vs. causation.

We made this change throughout the manuscript, replacing “promotes” and “drives” with “is linked to” and “is associated with”. 

3) Clarity regarding breeding vs. non-breeding/winter areas, timing of such, etc... e.g. as identified by Reviewer 2. To the extent that genetic structure is a consequence of gene flow, which of course is tied directly to mating, there needs to be better description of these colonies and where/when breeding takes place versus where/when they are relative to this when not breeding. Like Reviewer 2, I was confused. The reader is provided insufficient information to understand the situation.

We regret that we were not as clear as we could have been on this point. We have added a few sentences to the Methods section to clarify why we made the decision to examine the period from 1 November to 28/29 February [lines 355-358].

Throughout, replace the apostrophe in F'ST with a straight apostrophe (e.g. using insert symbol in Word).

We made this change throughout.

Figure 2a & 2b. Double check the ordering/naming of the colonies. To the extent, n is larger for MAT than TEU, these two, at least, apparently switched order between a & b. I also wondered about DE & PR. Be 100% sure that all names are associated with the correct genetic info in the STRUCTURE diagram.

We checked all of this carefully, and believe all of these issues have been addressed.

Also, Figure 2a & 2b. Both should look identical structurally given they are essentially part of the same figure: 2a currently has more space and an additional solid black line on the left. The region names and brackets also need to better match the associated breeding colonies. Currently, TOD looks to be part of Alaska, especially in 2b. The brackets and region names in general appear to be less carefully aligned/applied in 2b versus 2a. There's also no reason the bracket for Japan should extend well beyond the colony info. Making things exact is easy using PowerPoint by inserting figures, then adding text/brackets and aligning using grid lines. Seems to me 2a and b could have easily been pre-assembled into a single jpeg.

We have made extensive changes to the figures, following the recommendations of all reviewers. We believe they are now substantially improved.

Seems to me given their geographic location and their genetic similarity, SGG and TRI should be ordered together in all STRUCTURE figures/diagrams.

We respectfully disagree with this suggestion. The colony at TRI is closer to PI than it is to SGG, and similar to the distances between several other BC colonies. And while physically close, the two colonies are in quite different oceanographic zones – TRI in the transition zone between the California and Alaska Currents, but SGG in the Alaska Current. Finally, the PCoA provided some (weak) evidence that the colony at TRI is genetically distinct from all other colonies, as we concluded in our earlier (and less comprehensive) analysis (Abbott et al. 2014).

In general, bracket lines on all STRUCTURE figures should line up with the vertical bracket lines in the STRUCTURE diagram.

We believe these issues have now been addressed.

Figure 4. Relevance of colors and legend. Change label for Protection to PR.

We changed the label for PR, and added this sentence to the Figure caption: “The colours correspond to the three groups detected using STRUCTURE (Figs 2 and 3): western Pacific (yellow), the larger eastern Pacific group (purple), and smaller eastern Pacific group (blue).”

Figure 6. Correlation coefficients are generally designated using 'r' not 'p'. Until I read the text, I thought the figure included p-values of non significance. For clarity, replace 'p' with 'r'.

Those symbols were meant to be the lower case Greek letters rho, the symbol for Spearman’s correlation coefficient. But we have learned that this symbol is used as the coefficient for the entire population, and not for a sample. We have made this change in Fig. 6.

Line 393 and 394. Replace Table 4 with Table 3.

We made this change [lines 421-422].

Line 419. Replace 'Moore, Cleland' with 'Moore and Cleland'

We made this change [line 449].

Line 420. Identify which cluster

That should have read “cluster two” rather than “one cluster”. We made the change [line 450]. 

Line 436-440. Given that you're dealing with Q>60% throughout, use it once only... or at most twice. E.g. The eastern Pacific colonies split into two clusters at Q>60%. Cluster one included... etc. Cluster two included... In cluster one, delete 'Protection'. In both cases, precede the last colony in the list by 'and'. Why were SEF and MID singled out given all member of cluster 1 and cluster 2 were associated with the respective clusters at Q>60%?

We went through all of the STRUCTURE results very carefully, and think the text and Figure captions are now accurate and succinct [lines 441-481]. 

Line 447. Delete Protection. The text says 60%. Which is it 50 or 60%?

See comment above.

Line 448. Delete 'and' before Protection. Replace '&' with 'and' before Ano Nuevo. The text says 60%. Which is it 50 or 60%?

See comment above.

Line 520. Figure 6b says -0.21. Which is it?

We made this change - -0.22 is correct [line 554].

Lines 641-643. Given the amount of spread, i.e. low correlation, and fact the correlation was only significant at 0.05, I think this is a bit overstated.

We thought carefully about how to deal with the marginal correlation. In general, we believe a correlation at (two-tailed) P = 0.056 in the direction predicted by the hypothesis being tested should be considered statistically significant. Our resident biostatistician (M. Drever) also noted that given dispersion in the data, we are justified in using P = 0.1 to avoid Type II error. We added this to the Methods section [lines 387-388].

Reviewer #1 

371-372: negative FST values are an artifact of some programs and should be replaced by zeros.

We made this change to text and tables.

389-390: A significant Mantel’s test indicates correlation between genetic and physical distance but this doesn’t translate necessarily into isolation by distance. The fact that you get a significant Mantel’s test when you include populations from both sides of the Pacific but not when you analyze eastern Pacific colonies alone (lines 516-518) indicate that in the former the pattern is driven by strong pop structure between eastern and western Pacific colonies rather than by IBD. See Meirmans, P.G. 2012. The trouble with isolation by distance. Molecular Ecology 21: 2839-2846.

We agree with the reviewer, and thank her or him for the advice. We removed the test for IBD across all western and eastern North Pacific colonies. 

516-518: I would say that all results from Mantel’s should be presented together but testing for IBD between eastern and western Pacific colonies may be inappropriate (see above).

We agree, and have removed any testing of IBD across the full range.

547-548: See comments above. This is structure between two differentiated groups rather than IBD.

We agree.

Line 682: Analysis of highly differentiated loci in thick-billed murres (Tigano et al. 2017) has revealed a similar pattern, although not quantitively. The three Atlantic colonies that were more genetically similar at these outlier loci were also the colonies showing the highest overlap in wintering distributions. However, this pattern could have also been explained by colonization history following the retreat of the Pleistocene ice sheets or selection during the non-breeding time.

We added this result, and the Tigano et al. (2017) reference [lines 715-718]. 

Figure 1. Please show all colonies included in the study and highlight those that have also GLS data

We initially produced a figure showing all of the colonies, both in Japan and in North America. But this necessitated a figure of unacceptably large scale, creating issues around detail and visual separation of colonies. This was especially true of the western Pacific colonies - relative to the eastern Pacific, they are very close together. So, given that we did no geolocator tagging at the colonies in Japan for this study, that the focus of the work in Japan was to demonstrate the very strong east vs. west Pacific genetic split, we thought it better to go with a high quality map of the North American colonies rather than a low quality map showing all colonies. Locations of the sites in Japan are available in the Appendices, however. 

Figure 2-4 Structure plot image quality is low and pop labels are hard to read. Make sure that images are high quality.

We made many changes to the figures, and believe we have addressed the comments of all three reviewers.

Fig. 4 What do colors represent? Add legend.

We added a sentence to the figure caption to explain the relevance of the different colours [lines 499-500].

Fig. 6 Add p-values on graphs and/or figure caption.

We made these additions.

Reviewer #2

My only real issue with the paper is that the abstract, intro, and discussion use language that makes it appear the authors found a causal relationship when they only tested a correlation. The study is framed as a test of the hypothesis that segregation in the wintering distributions drives genetic differentiation among breeding colonies. However, the paper does not test a causal mechanisms by which nonbreeding segregation would drive genetic differentiation, and as a paper the authors cite (Rayner et al 2011) points out, “Spatial segregation during the non-breeding season appears to provide an intrinsic barrier to gene flow among seabird populations that otherwise occupy nearby or overlapping regions during breeding, but how this is achieved remains unclear.” What the study does do is test if there is a correlation between genetic differentiation and degree of difference in space use during the nonbreeding season. I think the language can easily be revised to clarify that this distinction by removing words like “promotes” and “drives” that imply causation and replacing them with words like “correlates with” that make it clear what the authors are testing.

We completely agree, and addressed this issue in replies to the Editor’s comments.

The authors only set up one hypothesis test in the intro (winter segregation is correlated with genetic structure), but they test an additional hypothesis in the paper (distance between breeding colonies is correlated with genetic structure). I assume the authors test this to evaluate if average dispersal difference accounts for genetic structure, assuming all populations have equal mean dispersal differences (which is not a given). Since the authors present results of both tests and use them to say that overwintering overlap better accounts for genetic difference than mean isolating distance, both hypothesis tests should be set up in the intro. 

We completely agree, and addressed this issue in comments to the Editor above

The authors also discuss two additional hypotheses in the discussion: genetic structure correlates with egg lay date or morphological differences like body size or mass (potentially indicating differences in foraging niche). These are also testable hypotheses, and it appears the authors have the data to test them instead of just discussing them. I think this could potentially strengthen the paper, though it’s not essential.

First, there is, as we discuss, a clear east-west difference in body size, but the larger birds from Asia do not interact or cohabitate with the smaller birds from North America - the two phenotypes are completely segregated throughout the year. As for structure being due to differences in timing of breeding, we did briefly test that hypothesis and concluded that it is unlikely to underlie the population genetic structuring in rhinoceros auklets, as it does in Cook’s petrel. We then provided the evidence we used to arrive at that conclusion - there is a latitudinal trend in timing of egg-laying (typically about one month later in the north than in the south, but with much variation), but the genetic variation does not match that latitudinal pattern [lines 698-705].

A limitation of the study is that the authors did not consider temporal segregation, only spatial segregation. Because of that, it is unclear to what degree these breeding colonies are potentially in contact during the nonbreeding season. The geolocator study cited from Japan (Takahashi et al. 2015) did find that space use varied with month, so I’m curious why the authors did not consider or account for that. I think the authors should address this in the discussion at a minimum.

We addressed this issue in comments to the Editor above.

Authors use inconsistent terminology for geolocators. I found the following terms used in the ms: geolocator, geologger, logger, and GLS device. Bird carrying geolocators were also referred to as loggered and geologgered. Best to pick one and use it consistently to avoid confusion.

We have addressed this issue – after initially introducing “light-level geolocator (GLS) tags”, we refer in the rest of the manuscript to “geolocator tags”, “tags” and “tagging”

325: When do the colonies initiate and end breeding? How does that relate to the “winter” dates selected?

We addressed this issue in replies to the Editor.

365, comma missing

We added the comma.

631-633: I found this sentence confusing. I recommend “in which the population showed the unusual behavior of remaining in close proximity to a breeding colony year-round and foraging…”

That amended that sentence [lines 665-667].

640-643: change word from “promotes” as you don’t have evidence of a mechanistic relationship.

We addressed this issue, as outlined in the replies to the Editor’s comments.

658: not “driven,” correlated with. Again, no causation has been shown.

We addressed this issue, as outlined in the replies to the Editor’s comments.

669-670: These are alternative hypotheses that are testable.

We addressed this issue as described above [lines 698-705].

Table 4: I think there’s an error in Middleton, which shows more track files recovered than geolocators recovered.

We corrected the error.

Fig 1: As this is the only full study area map, the Western Pacific sites should be included and differentiated with a different color or icon.

We addressed this issue in response to reviewer #1. The full map is in the Appendices.

Fig 4: Unclear what the colors represent. Please include a color legend. It would be logical for colors to be grouped by the five regions identified in figures 2 and 3, but there are more than five colors presented here. Please clarify.

We addressed this issue in response to reviewer #1.

Figure 5: The resolution of the file attached for figure 5 is low, so it’s hard to figure out what the gray lines are. The caption says they represent the geolocator tracks, but the look like dashed, horizontal lines, whereas geolocator tracks are typically presented as a single connected line. The gray lines here look more like shading, but it’s unclear how that shading was generated. Does this incorporate some metric of uncertainty around each point in the track or aggregate all tracks in some way? Please clarify.

Figure 5 is now provided at an increased resolution of 800 ppi. The TwilightFree algorithm provides the most likely daily point locations on a pre-determined grid of 1°×1° cell sizes, based on the each bird’s light record for that day. These underlying grid makes the points appear as lines, but they are point locations of all birds from each colony, with no uncertainty depicted. We provided those points to show the entire wintering distributions of all birds for each colony. The Utilization Distribution (UD) polygons are then derived from these points.

---

## [Editor Report · Decision Letter 1]

18 Sep 2020

Geolocator tagging links distributions in the non-breeding season to population genetic structure in a sentinel North Pacific seabird

PONE-D-20-09231R1

Dear Dr. Hipfner,

We’re pleased to inform you that your manuscript has been judged scientifically suitable for publication and will be formally accepted for publication once it meets all outstanding technical requirements.

Kind regards,

Janice L. Bossart

Academic Editor

PLOS ONE
---

## [Editor Report · Acceptance letter]

26 Oct 2020

PONE-D-20-09231R1 

Geolocator tagging links distributions in the non-breeding season to population genetic structure in a sentinel North Pacific seabird 

Dear Dr. Hipfner:

I'm pleased to inform you that your manuscript has been deemed suitable for publication in PLOS ONE. Congratulations! Your manuscript is now with our production department. 

Kind regards, 

on behalf of

Dr. Janice L. Bossart 

Academic Editor

PLOS ONE